# Single-cell transcriptomics reveal the dynamic of haematopoietic stem cell production in the aorta

Chloé S. Baron[1], Lennart Kester[1], Anna Klaus[1], Jean-Charles Boisset[1], Roshana Thambyrajah[2], Laurent Yvernogeau[1], Valérie Kouskoff[3], Georges Lacaud[2], Alexander van Oudenaarden[1] & Catherine Robin[1,4]

Haematopoietic stem cells (HSCs) are generated from haemogenic endothelial (HE) cells via the formation of intra-aortic haematopoietic clusters (IAHCs) in vertebrate embryos. The molecular events controlling endothelial specification, endothelial-to-haematopoietic transition (EHT) and IAHC formation, as it occurs in vivo inside the aorta, are still poorly understood. To gain insight in these processes, we performed single-cell RNA-sequencing of non-HE cells, HE cells, cells undergoing EHT, IAHC cells, and whole IAHCs isolated from mouse embryo aortas. Our analysis identified the genes and transcription factor networks activated during the endothelial-to-haematopoietic switch and IAHC cell maturation toward an HSC fate. Our study provides an unprecedented complete resource to study in depth HSC generation in vivo. It will pave the way for improving HSC production in vitro to address the growing need for tailor-made HSCs to treat patients with blood-related disorders.

[1] Hubrecht Institute-KNAW, University Medical Center Utrecht, Uppsalalaan 8, 3584 CT Utrecht, The Netherlands. [2] CRUK Stem Cell Biology Group, Cancer Research UK Manchester Institute, The University of Manchester, Aderley Park, Aderley Edge, Macclesfield SK10 4TG, UK. [3] Division of Developmental Biology and Medicine, The University of Manchester, Michael Smith Building, Oxford Road, Manchester M13 9PT, UK. [4] Regenerative Medicine Center, University Medical Center Utrecht, 3584 EA Utrecht, The Netherlands. These authors contributed equally: Chloé S. Baron, Lennart Kester. Correspondence and requests for materials should be addressed to C.R. (email: c.robin@hubrecht.eu)

Haematopoietic stem cells (HSCs) produce billions of blood cells every day throughout life, owing to their multi-potency and self-renewal properties. In the clinic, HSC transplantations are common practice to treat patients with blood-related genetic disorders and malignancies. However, finding match donor HSCs for the increasing number of transplantations has become an issue. Intensive years of research have focused on the possibility to generate HSCs in vitro that would serve as a potential alternative source for these life-saving cells. An unlimited access to in vitro patient-derived HSCs would also facilitate drug screening and allow studying the development of blood-related diseases such as leukemia. The fundamental finding that all HSCs derive from "haemogenic" endothelial cells during embryonic development has paved the way to recent advancements in the generation of transplantable HSCs in vitro[1–4]. However, the molecular mechanism of the endothelial specification and its conversion into HSCs as it occurs in vivo in the course of embryonic life is still poorly understood. Such knowledge would certainly help to improve the production of bona fide transgene-free HSCs, which remains the optimal choice for therapies.

During mouse embryonic development, HSCs are first detected in the main arteries (such as the aorta of the aorta–gonad–mesonephros (AGM) region), starting at embryonic day (E)10.5, as shown by long-term in vivo transplantation assays[5–7]. HSCs reside in intra-aortic haematopoietic clusters (IAHCs) attached to the wall of the aorta between E9.5 and E14[8,9]. IAHCs are found in the ventral side of the aorta in most vertebrate species, with the exception of the mouse where low numbers of IAHCs are also present in the dorsal side[10]. IAHCs express haematopoietic stem and progenitor cell (HSPC) markers (e.g., c-kit, CD41)[11–13] and are completely absent in mouse models devoid of HSCs (e.g., $Runx1^{-/-}$ mice)[14–16]. Beside haematopoietic characteristics, IAHCs also express endothelial markers (e.g., CD31 (PECAM1), VE−Cadherin (CDH5, CD144), Tie2, CD34)[8,9,17], which supported a presumptive endothelial origin[18]. This has since been proven in various animal models by performing live confocal imaging and in vivo/in vitro lineage tracing experiments[12,16,19–24]. Altogether these studies established that HSPCs and IAHCs are generated from a transient subset of endothelial cells, named haemogenic endothelial (HE) cells, through an endothelial-to-haematopoietic transition (EHT). Although numerous IAHC cells (~700 IAHC cells at E11) are present in the aorta[9], they contain only very few HSCs (~1−3 HSCs/AGM)[25,26]. IAHCs also contain very few committed progenitors (~22 progenitors/AGM). Most of them are indeed in the circulating blood, coming from the yolk sac (YS)[8,27,28]. Finally, IAHCs also contain few HSC precursors (~12–18 pre-HSCs/AGM, capable of long-term multilineage reconstitution upon transplantation in newborn recipients or after a step of in vitro culture on stromal cell line prior transplantation)[8,29]. In fact, most IAHC cells are probably very immature pre-HSCs (with no transplantable capacity yet) that progressively mature toward an HSC fate. Accordingly, numerous type I pre-HSCs (CDH5+CD45−) able to mature into type II pre-HSCs (CDH5+CD45+) and then into functional HSCs have been identified in an in vitro AGM dissociated/reaggregated culture system[30,31]. Since the number of E11.5 AGM pre-HSCs correlates with E12.5 fetal liver HSCs, it strongly suggests that IAHC cells form the adult HSC pool after migration and maturation in the fetal liver[32]. However, the exact cell composition of IAHCs in vivo, their putative heterogeneity, and the molecular steps leading to the generation of IAHCs and HSCs from the endothelium remain poorly understood. A novel approach that does not rely on functional in vitro/in vivo assays would need to be deployed to answer these questions.

In this study, we identified the molecular changes occurring during endothelial specification, EHT and IAHC formation, by using single-cell mRNA-sequencing (scRNA-Seq). This technique allows us to in silico purify IAHC cells, therefore removing any contamination from the dataset. We established that IAHCs are composed of pre-HSCs (type I and II) and committed progenitors, discernible by opposing gradients of endothelial and haematopoietic transcripts. We identified the genes and transcription factor (TF) networks activated for the silencing of the endothelial program and the initiation of the haematopoietic program during EHT as it occurs in vivo in the aorta of E10 and E11 embryos. Finally, we developed a new technique to isolate single whole IAHCs (wIAHCs) in the ventral and dorsal side of the aorta and demonstrated that they have similar transcriptomes, at both E10 and E11. Overall, we provide an unprecedented complete resource to study in depth the molecular changes occurring in the successive steps leading to HSC formation.

## Results

**scRNA-Seq allows in silico purification of IAHC cells.** To determine which staining procedure allows IAHC cell isolation to the highest purity, several approaches were compared (Supplementary Fig. 1). Cells were stained with anti-c-kit antibodies alone or in combination with anti-CD31 antibodies, either after dissociation of the whole AGM (total staining, TS)[9] or directly inside the aorta prior to AGM dissection (intra-aorta staining, IAS)[8,12,33]. Single c-kit+ cells (TS, Supplementary Fig. 1a; IAS, Supplementary Fig. 1b) or c-kit+ cells with CD31 index recording (IAS; Supplementary Fig. 1c) were sorted. Besides composing IAHCs (Supplementary Fig. 1d, arrow heads), c-kit+ cells are also present in the aorta surrounding mesenchyme (Supplementary Fig. 1d, asterisks)[9,12]. To evaluate the possible contamination by non-IAHC c-kit+ cells, anti-c-kit-PE antibodies were injected inside the aorta, AGMs were then dissected, dissociated, and cells were stained with APC-anti-c-kit, FITC-anti-CD31, and anti-CD45 antibodies. Single non-IAHC cells were then sorted (c-kit-PE−c-kit-APC+CD31−CD45− cells, thereafter referred to as c-kit+ cells, mesenchyme staining or MS; Supplementary Fig. 1e).

scRNA-Seq was performed on a total of 542 c-kit+ (TS, IAS, MS), c-kit+CD31+, and c-kit+CD31− (IAS) cells using CEL-Seq[34,35] (Fig. 1a). RaceID[34], an algorithm developed for rare cell type identification in complex populations of single cells, identified six main clusters (Fig. 1b). Cells had varying levels of *Kit*, *Pecam1* (*CD31*), and *Ptprc* (*CD45*) mRNA expression (Fig. 1c, and Supplementary Fig. 1f). Part of the c-kit+ cells (IAS) and most c-kit+CD31+ cells (IAS) were in clusters 4 and 5 (Fig. 1b). They expressed known IAHC markers such as *Gpr56*, *CD34*, *Runx1*, and *Gata2*[12,15,36,37] (Fig. 1d). Noteworthy, some cells in clusters 4 and 5 did not express IAHC markers but markers for haematopoietic cells located in sub-aortic patches such as *Gata3* and *Cdkn1c* (*P57*, *Kip2*)[38,39] (Supplementary Fig. 2a, b). In contrast, the other part of c-kit+ cells (IAS) and most c-kit+ CD31− cells (IAS) and c-kit+ cells (TS) clustered together with the c-kit+ cells (MS), forming the clusters 1, 2, 3, and 6 (Fig. 1b). These cells did not express IAHC markers (Fig. 1d). Cells in clusters 1, 2, and 3 expressed genes related to mesenchymal lineages[40] (e.g., *Crabp1*, *Cxcl12*, *Pdgfrb*, *Sox11*, *Ptprd*, *Col3a1*, and *Epha7*; Supplementary Fig. 2c–i). Accordingly, they were located in the mesenchyme in close proximity to the aortic endothelium, as shown by single-molecule (sm)RNA-FISH (e.g., *Epha7*; Supplementary Fig. 2j). Cluster 6 expressed genes related to erythro-myeloid progenitors (EMPs)[41–43] (e.g., cells expressing *Csf1r* and *Fcgr3* (*CD16*) but not *Myb* transcripts; Supplementary Fig. 2k–m).

The optimal IAHC cell purity was obtained after IAS based on c-kit and CD31 expression (97% of c-kit+CD31+ cells expressed

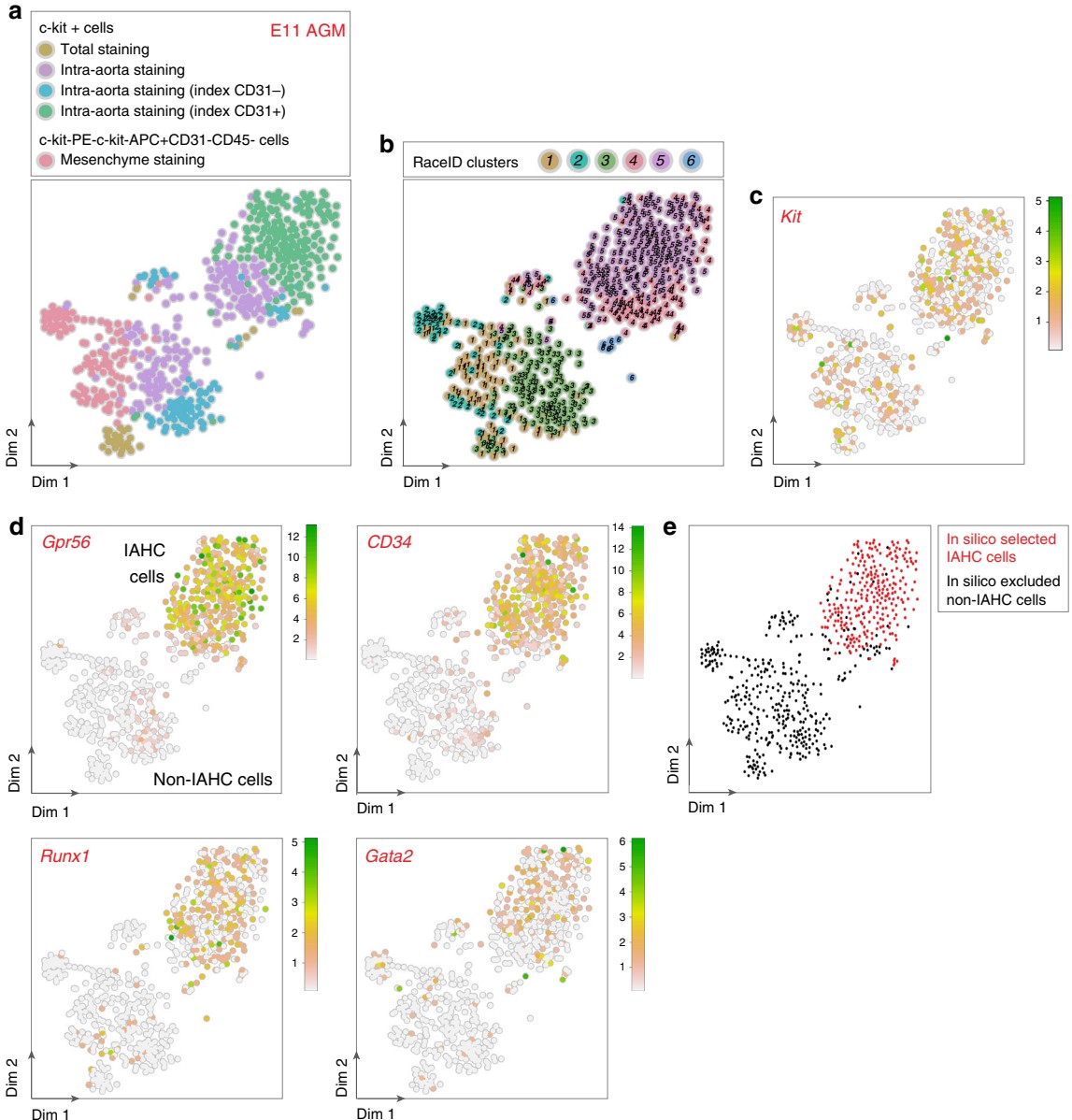

**Fig. 1** scRNA-Seq allows in silico purification of IAHC cells from E11 AGM. **a–d** t-SNE maps displaying as colored dots 542 single cells isolated from the aorta–gonad–mesonephros (AGMs) region of E11 embryos. **a** t-SNE map displaying 37 c-kit$^+$ cells sorted after total staining (brown dots), 215 c-kit$^+$ cells sorted after intra-aorta staining (purple dots), c-kit$^+$ cells sorted with CD31 fluorescence intensity index after intra-aorta staining (92 c-kit$^+$CD31$^-$ cells, blue dots; 198 c-kit$^+$CD31$^+$ cells, green dots), and 114 c-kit$^-$PE$^-$c-kit$^-$APC$^+$CD31$^-$CD45$^-$ cells (pink dots). **b** t-SNE map displaying single cells from **a** in clusters identified after RaceID analysis. Different numbers and colors highlight the different RaceID clusters. **c, d** Expression of (**c**) *Kit* and (**d**) *Gpr56*, *CD34*, *Runx1*, and *Gata2* marker genes projected on t-SNE maps. Color bars, number of transcripts. Dim dimension. **e** t-SNE map displaying in silico selected IAHC cells (in red) and excluded non-IAHC cells (in black)

**Table 1 Percentages of IAHC cells (identified by *Gpr56* expression) after different antibody staining and cell sorting strategies**

| Sorted cells (staining strategy) | Percentage of cells with *Gpr56* transcripts |
|---|---|
| c-kit$^+$ cells (total staining) | 6 |
| c-kit$^+$ cells (intra-aorta staining) | 53 |
| c-kit$^+$CD31$^-$ (intra-aorta staining) | 25 |
| c-kit$^+$CD31$^+$ (intra-aorta staining) | 97 |
| c-kit-PE$^-$c-kit-APC$^+$CD31$^-$CD45$^-$ (mesenchyme staining) | 0 |

*Gpr56* transcripts; Fig. 1d, Table 1). However, 25% of IAHC cells (*Gpr56*$^+$) would be lost in the c-kit$^+$CD31$^-$ fraction after sorting of the c-kit$^+$CD31$^+$ fraction (Table 1). Therefore, it is technically difficult to isolate all IAHC cells as a pure population due to unavoidable limitations related to antibody staining procedure, the difficulty to set up optimal sorting gates and cell sorting errors. However, scRNA-Seq represents a powerful tool to correct for cell contamination by in silico purifying IAHC cells based on specific marker gene expression. To study pure IAHC cells, we therefore selected the cells that had more than one *Gpr56* transcript and filtered out the cells that had more than two transcripts of one or more of the non-IAHC genes (Fig. 1e).

**Isolation of successive cell populations during EHT**. To analyze the molecular steps leading to IAHC formation, different cell populations were sorted based on the differential expression of the transcriptional repressors Gfi1 and Gfi1b, and c-kit[12,44]. Single non-HE cells (Cdh5$^+$Gfi1$^-$Gfi1b$^-$c-kit$^-$), HE cells (Cdh5$^+$ Gfi1$^+$Gfi1b$^-$c-kit$^-$), and cells undergoing EHT (Cdh5$^+$Gfi1$^+$ Gfi1b$^-$c-kit$^+$) were sorted from E11 and E10 AGMs and sequenced (Supplementary Fig. 3a, b, respectively). In addition, sorted E11 and E10 IAHC cells were in silico purified (Fig. 1e). Finally, type I pre-HSCs (c-kit$^+$Cdh5$^+$CD45$^-$) and type II

pre-HSCs (c-kit$^+$Cdh5$^+$CD45$^+$) were sorted from E11 AGMs (Supplementary Fig. 3c). For comparison, c-kit$^+$ HSPCs were also sorted from E11 and E10 YSs (Supplementary Fig. 3d, e, respectively). The transcriptomes of the single cells from the different populations were then compared to each other at E11 and E10 (Figs. 2a, 3a, left panels, respectively).

The Monocle algorithm[45] was used to arrange the single-cell expression profiles on a developmental axis. Both at E11 and E10, the different populations aligned along a pseudotime axis that followed the successive steps of the in vivo EHT process

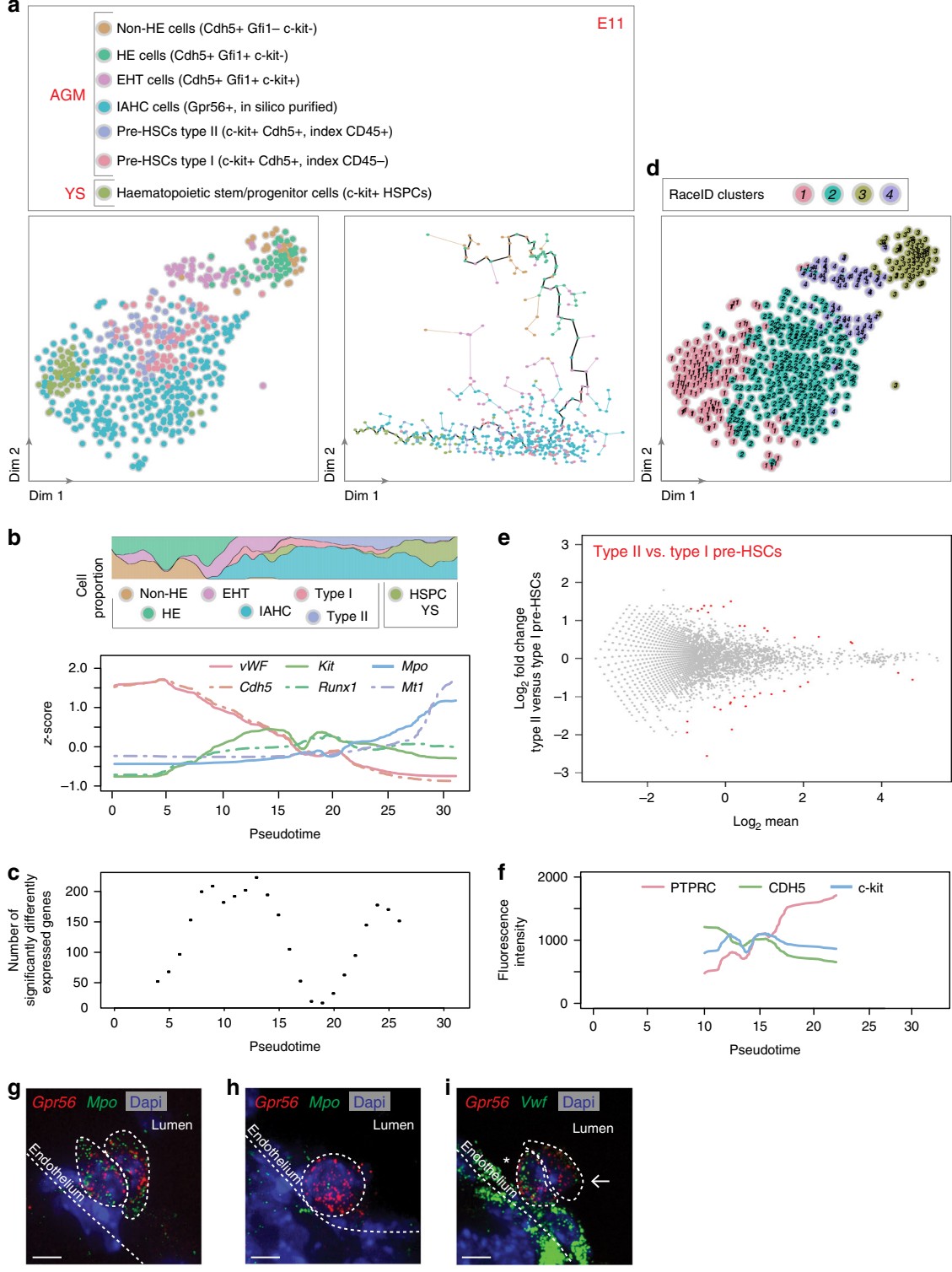

(Figs. 2a, 3a, right panels, respectively). The concomitant loss of the endothelial program and initiation of the haematopoietic program during developmental progression was illustrated by the decreased expression of *Vwf* and *Cdh5* (endothelial marker genes) and the increased expression of *Kit*, *Runx1*, *Mpo*, and *Mt1* (haematopoietic marker genes) (Fig. 2b (E11) and Fig. 3b (E10)). The most extensive transcriptome remodeling occurred during EHT, where over 250 genes significantly changed their expression (Fig. 2c). For example, endothelial cells highly expressed gap junction transcripts (e.g., *Gja4* and *Gja5*, Supplementary Fig. 4a, b) that were downregulated when cells underwent EHT. HE cells expressed *Prom1*[46], *Sox17*, and *Gfi1* (Supplementary Fig. 4c–e), while *Kit*, *Runx1*, *Sfpi1*, and *Gpr56* expression increased as cells underwent EHT (Supplementary Fig. 4f–i) and remained expressed in IAHC cells. The proliferation of IAHC cells was recently reported[32,47]. Using the KEGG database, we analyzed the "cell cycle" pathway according to pseudotime and indeed detected many genes of this pathway in most cell populations of our E11 dataset (Supplementary Fig. 5). Importantly, the gene expression observed in the different populations (as reported above) did not change after correction for cell cycle and proliferation genes. Overall, our analysis validated the identity of the various sorted cell populations chosen based on the expression of expected marker genes. Furthermore, this dataset represents a unique and reliable resource to examine gene expression changes during the development of endothelial cells into IAHC cells.

**IAHCs contain pre-HSCs and progenitors at E11 and E10.** RaceID analysis, performed to identify the different cell types present in IAHCs, divided IAHC cells into two main clusters at E11 (Fig. 2d; clusters 1 and 2). Part of IAHC cells clustered with YS HSPCs (cluster 1) while the other part clustered with sorted pre-HSCs (cluster 2). Most cells in cluster 1 expressed haematopoietic markers such as *Rac2* and *Mpo* (Supplementary Fig. 6a, b) and some cells also expressed committed erythroid lineage transcripts (e.g. *Klf1*, *Gata1*, and *Mt1*; Supplementary Fig. 6c–e). On the other hand, cells in cluster 2 expressed endothelial transcripts (e.g., *Vwf*, *Cdh5*, *Pecam1* (CD31), *Tie1*, and *Eng* (CD105))[17,48,49] (Supplementary Fig. 6f–j) and *Procr* (CD201), a gene recently described as a type I pre-HSC marker[29] that we found also highly expressed in EHT and HE cells (Supplementary Fig. 6k). This suggests the presence of two populations in IAHCs at E11, namely, committed haematopoietic progenitors and pre-HSCs, which are discernible by opposing endothelial and haematopoietic signatures (Supplementary Fig. 6l). RaceID analysis also identified these two populations in IAHCs at E10 (clusters 1 (pre-HSCs) and cluster 2 (committed progenitors); Fig. 3c, d). To determine whether these populations are similar at E11 and E10, RaceID was performed on the combined dataset

(Supplementary Fig. 7a, b). This showed that both E10 and E11 committed progenitors (cluster 1, cells expressing *Rac2*, Supplementary Fig. 7b, c) and E10 and E11 pre-HSCs (cluster 2, cells expressing *Cdh5*, Supplementary Fig. 7b, d) clustered together. Overall, progenitors and pre-HSCs are already present in IAHCs at E10, with a similar transcriptome than at E11.

Our scRNA-Seq data were further analyzed to examine the transcriptome of type I and type II pre-HSCs in E11 IAHCs. Interestingly, the two populations clustered together (cluster 2, Fig. 2d). DE-Seq analysis[50] revealed that only 36 genes (out of 18,735 genes) were significantly differentially expressed between type I and type II pre-HSCs (Fig. 2e; Supplementary Data 1), confirming that phenotypically defined pre-HSCs (based on CD45 differential expression) are very similar at the transcriptome level in vivo. We then looked at the fluorescence intensities recorded during flow cytometry sorting of type I and type II pre-HSCs along the pseudotime axis (Fig. 2f). Although *Kit* expression was stable, a decreased expression of CDH5 and increased expression of PTPRC (CD45) were observed along the pseudotime axis. This therefore further validates the in vivo developmental order of events, with the maturation of type I pre-HSCs into type II pre-HSCs within IAHCs in vivo. Flow cytometry analysis also confirmed the lower abundance of phenotypically defined type I pre-HSCs at E11 compared to E10 (Supplementary Fig. 3c, f, respectively).

Zhou and colleagues recently performed scRNA-Seq analysis on four populations sorted from E11 AGM based on CD31 and CD45 but also CD41 and CD201 expression (according to Zhou et al. nomenclature: t1 CD201$^{low}$ [CD31$^+$CD45$^-$CD41$^{low}$ c-kit$^+$CD201$^{low}$]; t1 CD201$^{high}$ [CD31$^+$CD45$^-$CD41$^{low}$c-kit$^+$ CD201$^{high}$]; t2 CD41$^{low}$ [CD31$^+$CD45$^+$CD41$^{low}$]; t2 CD201$^{high}$ [CD31$^+$CD45$^+$c-kit$^+$CD201$^{high}$])[29]. We compared the genes differentially expressed between our cell populations (Fig. 2a) to the genes differentially expressed between their four populations. The analysis confirmed the identity of our pre-HSCs type I, which had a significant high overlap with the t1 CD201$^{high}$ population (enriched in pre-HSCs type I) (Supplementary Fig. 8a [left panel] and 8b [42% of genes in common]), but not with the t2 CD41$^{low}$ population (enriched in pre-HSCs type II)[29] (Supplementary Fig. 8a [left panel] and 8c [7% of genes in common]). The analysis also confirmed the identity of our pre-HSCs type II, which had a significant overlap with both t1 CD201$^{high}$ (although lower than with pre-HSCs type I) (Supplementary Fig. 8a [right panel] and 8d [27% of genes in common]) and t2 CD41$^{low}$ populations (Supplementary Fig. 8a [right panel] and 8e [21% of genes in common]). Accordingly, t1 CD201$^{high}$ cells (similar to our pre-HSCs type I) expressed *Gpr56*, *Pecam1*, and *Kit* but not yet *Ptprc* (Supplementary Fig. 8f). t2 CD41$^{low}$ cells (similar to our pre-HSCs type II) expressed *Gpr56*, *Pecam1*, *Kit*, and *Ptprc* (Supplementary Fig. 8f). As expected, our pre-HSCs had a

**Fig. 2** Pre-HSCs and progenitors are discernible by opposing gradients of endothelial and haematopoietic transcripts in E11 IAHCs. **a** t-SNE map of 554 single cells isolated from E11 embryo AGM and yolk sac (YS). **a** (left panel) t-SNE map displaying 27 sorted non-haemogenic endothelial cells (non-HE cells, Cdh5$^+$Gfi1$^-$c-kit$^-$, brown dots), 45 haemogenic endothelial cells (HE cells, Cdh5$^+$Gfi1$^+$c-kit$^-$, green dots), 43 cells undergoing endothelial-to-haematopoietic transition (EHT cells, Cdh5$^+$Gfi1$^+$c-kit$^+$, purple dots), 282 IAHC cells (turquoise dots; c-kit$^+$ cells in silico purified based on *Gpr56* expression from Fig. 1a), and HSC precursors (58 pre-HSCs type I [c-kit$^+$Cdh5$^+$, index CD45$^-$, pink dots] and 55 type II [c-kit$^+$Cdh5$^+$, index CD45$^+$, violet dots]) from AGMs. 44 c-kit$^+$ cells were also sorted from YS (haematopoietic stem and progenitor cells, HSPCs, khaki dots). **a** (right panel) Pseudotime analysis by Monocle algorithm of the cells shown in **a** (left panel) (same color code). **b** Top panel, proportion of each cell type shown in **a** along pseudotime; Bottom panel, endothelial and haematopoietic marker gene expression along pseudotime (*vWF* [von Willebrand factor], *Cdh5* [VE-cadherin], *Kit*, *Runx1*, *Mpo* [Myeloperoxidase], *Mt1* [Metallothionein 1]). **c** Number of significantly differentially expressed genes along pseudotime. **d** t-SNE plot displaying the single cells shown in **a** in clusters identified by RaceID analysis. **e** DE-Seq analysis showing the differential gene expression between type I and type II pre-HSCs (36 genes were significantly differentially expressed, red dots). **f** Fluorescence intensity of markers used to sort pre-HSCs along pseudotime (PTPRC [CD45], CDH5, c-kit). **g–i** E11 embryo cryosections hybridized with smFISH probes against *Gpr56* (red dots) (**g–i**) and *Mpo* (green dots) (**g**, **h**) or *Vwf* (green dots) (**i**). IAHC cells (*Gpr56*$^+$) expressing high (asterisk) or low/no levels (arrow) of *Vwf* in (**i**). Dash lines delimitate the position of the aortic endothelium and IAHC cells. Dapi marks the nuclei (blue)

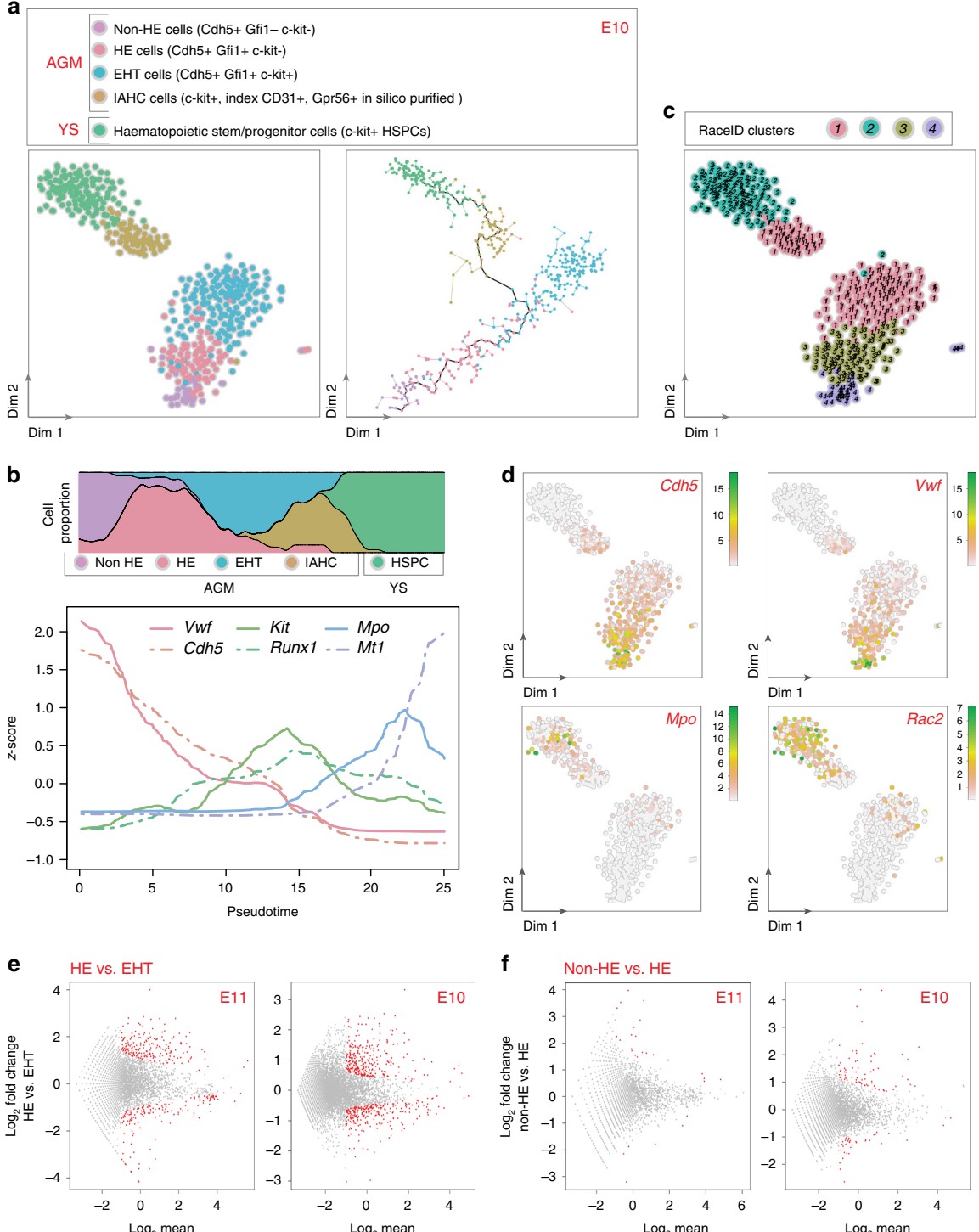

**Fig. 3** Non-HE and HE cells have different transcriptome at E10. **a** t-SNE map displaying 464 single cells isolated from E10 embryo AGM. **a** (left panel) t-SNE map displayed 39 sorted non-hemogenic endothelial (HE) cells (purple dots), 96 HE cells (pink dots), 116 EHT cells (blue dots), and 73 IAHC cells (in silico purified on Gpr56 expression; brown dots) from E10 AGM, and 140 HSPCs from E10 YS (green dots). **a** (right panel) Pseudotime analysis by Monocle algorithm of the sorted cells shown in **a** (left panel) (same color code). **b** Top panel, proportion of each cell type shown in **a** along pseudotime; Bottom panel, endothelial and haematopoietic marker gene expression along pseudotime (*Vwf, Cdh5, Kit, Runx1, Mpo, Mt1*). **c** t-SNE plot displaying single cells from **a** in clusters identified by RaceID analysis. Different numbers and colors highlight different RaceID clusters. **d** Expression of *Cdh5, Vwf, Mpo*, and *Rac2* marker genes projected on t-SNE maps. Color bars, number of transcripts. Dim dimension. **e** DE-Seq analysis showing the differential gene expression between HE and EHT cells at E11 (**e**, left panel, 370 genes significantly differentially expressed, red dots) and E10 (**e**, right panel, 572 genes). **f** DE-Seq analysis showing the differential gene expression between non-HE and HE cells at E11 (**f**, left panel, 30 genes) and E10 (**f**, right panel, 109 genes). Non-significantly differentially expressed genes, gray dots

significant overlap with AGM t2 CD201[high], E12 fetal liver (Lin[−] Sca1[+]Mac1[low]CD201[+]), and adult BM HSCs[29] (Supplementary Fig. 8a). Of note, Zhou and colleagues mentioned that the t2 CD41[low] population was contaminated by committed progenitors[29]. Indeed, this population had a significant high overlap with our progenitor cells (Supplementary Fig. 8g, h [21% of genes in common]). To further confirm the committed progenitor identity of cluster 1 cells (Fig. 2d), four populations were sorted from E11 AGMs and tested in clonogenic assays (c-kit[+]CD31[+]Cdh5[−] CD45[+] and c-kit[+]CD31[+]Cdh5[−]CD45[−] cells (both in cluster 1), and c-kit[+]CD31[+]Cdh5[+]CD45[+] and c-kit[+]CD31[+]Cdh5[+]CD45[−] cells (both in cluster 2)). All fractions produced erythroid and/or myeloid colonies (Supplementary Fig. 8i; $n = 6$ independent experiments). However, such progenitors were highly enriched in the c-kit[+]CD31[+]Cdh5[−]CD45[+] fraction ($29.9 \pm 7.4$ progenitors/ 100 sorted cells) (Supplementary Fig. 8j), therefore confirming that cluster 1 encompass the committed progenitors. The t1 CD201[low] population, shown to have no transplantation potential[29], could be identified as non-IAHC cells since they highly overlap with this population (Supplementary Fig. 8k, l [28% of genes in common]) but not with our pre-HSCs (Supplementary Fig. 8a). Accordingly, t1 CD201[low] cells did not express Gpr56 and Ptprc (Supplementary Fig. 8f). Altogether, our data and comparative analysis confirmed the committed progenitor identity of cluster 1 cells and the pre-HSC identity of cluster 2 cells at E11. Moreover, our datasets provide complementary information and the possibility to perform comparative analyses with previously published datasets[29], independently of the markers used for cell sorting strategy.

smRNA-FISH for Gpr56 and Mpo or Vwf was performed on E11 embryo cryosections to localize committed progenitors and pre-HSCs in IAHCs, respectively. Cells with high amounts of Mpo transcripts (Fig. 2g), corresponding to committed progenitors, or with few to no Mpo transcripts, corresponding to pre-HSCs (Fig. 2h), were visible in Gpr56[+] IAHCs. As expected, Vwf transcripts were abundantly detected in aortic endothelial cells[17] (Fig. 2i). Similar to Mpo, the expression level of Vwf in Gpr56[+] IAHC cells was variable, with cells expressing Vwf transcripts being closer to the endothelium (Fig. 2i, asterisk). Overall, our analyses demonstrated that IAHCs are very similar at E10 and E11, containing pre-HSCs and committed progenitors expressing opposing gradients of endothelial (upregulated in pre-HSCs) and haematopoietic (upregulated in committed progenitors) transcripts.

**Non-HE and HE cells have very similar transcriptomes**. To further analyze the first steps leading to IAHC formation, non-HE, HE, and EHT cells were compared at both E11 and E10. RaceID analysis clustered most EHT cells together, which allows studying genes specifically expressed in this population evolving from an endothelial to a haematopoietic state. DE-Seq analysis revealed that fewer genes were differentially expressed between HE and EHT cells at E11 (370 genes) than at E10 (572 genes) (Fig. 3e; Supplementary Data 2 and 3). Interestingly, some pre-HSCs and HE cells clustered with EHT cells at both E11 (cluster 4, Fig. 2d) and E10 (cluster 1, Fig. 3c), reflecting the continuum of differentiation occurring between these successive related populations.

RaceID analysis clustered non-HE and HE cells together at E11 (cluster 3, Fig. 2d) but not at E10 (clusters 4 and 3, respectively, Fig. 3c). However, DE-Seq analysis showed that very few genes were differentially expressed between non-HE and HE cells at both time points (30 and 109 genes at E11 and E10, respectively; Fig. 3f, Supplementary Data 4 and 5). Nevertheless, some genes were nicely significantly differentially expressed in between non-

HE, HE, EHT cells, and IAHC cells. For example, at both E10 and E11, non-HE cells expressed Mgp while HE cells did not (Supplementary Fig. 9a). Furthermore, non-HE and HE cells expressed Epas1 and Id3 transcripts while EHT cells did not (Supplementary Fig. 9b, c). On the other hand, only EHT cells expressed Mycn transcripts (Supplementary Fig. 9d). To reinforce the relevance of the genes highlighted in our study, we performed different immunostaining techniques on AGM cells from E10 or E11 embryos for a selection of three new markers. IKZF2 (Helios), PROM1 (CD133), and GJA5 (Connexin 40) were chosen for their differential expression in our populations of interest (endothelium vs. IAHC cells) and for the commercial availability of antibodies. As predicted by our scRNA-Seq data, IKZF2 was much highly expressed at the protein level in c-kit[+]CD31[+] IAHC cells (86.6%) than in endothelial cells (15.8%), as shown by flow cytometry analysis (Supplementary Fig. 10a, b). On the other hand, PROM1 and GJA5 were expressed by CD31[+] endothelial cells but not by IAHC cells (c-kit[+]CD31[+] or Runx1[+] CD31[+] cells, Supplementary Fig. 10c–g [immunostaining on E11 live thick embryo slices] and Supplementary Fig. 10h–l [immunostaining on E10 embryo cryosections], respectively). Therefore, there is a good correlation between scRNA-Seq data and protein expression, which consolidates our datasets. Such datasets will be useful to study the functionality of specific genes of interest in specific populations of the aorta or to test their usefulness as new surface markers for specific cell isolation in the future.

To identify the pathways active in the aortic populations, we performed a KEGG pathway analysis on our E11 dataset. As expected, specific genes related to the BMP pathway (e.g., Bmp4, Bmpr1a, Bmpr2, Smad 6, Smad 7), Notch pathway (e.g., Dll1, Dll4, Notch1, Notch 3, Notch 4, Jag1), and mTORC2 pathway (e.g., Rictor, mTor, Deptor, Tti1) were expressed. We then examined in more details the KEGG cytokine pathway. A heatmap was generated where genes present in each cytokine pathway (according to KEGG analysis) were on the y-axis and E11 single cells (ordered along the pseudotime axis) were on the x-axis (Supplementary Fig. 11a). It revealed the dynamic expression of some known important regulators, including c-kit[51] and IL-3 (through IL-3R)[26]. Three groups of cytokines/growth factors were more active in endothelium, IAHC cells or committed progenitors. A KEGG pathway analysis performed on these three datasets highlighted the involvement of interleukins, TGF, chemokines, and/or BMP in the endothelium and IAHC cells (Supplementary Fig. 11b, c). Pathways involving hemostasis and cell surface interactions at the vascular wall were active in the committed progenitor group, possibly reflecting the detachment of these cells to the circulation (Supplementary Fig. 11d). Molecules involved in the immune system/inflammation were also active in the three groups (i.e., TNF, IL-6), in accordance to the literature where the inflammatory process was recently reported as an important key regulator of HSC development[52]. Our datasets will be useful to investigate in the future the requirement for specific cytokines and growth factors at specific time points during endothelial specification and HSC characteristic acquisition.

**TF network involved in IAHC formation at E11 and E10**. To get insight in the transcriptional program involved in EHT and IAHC formation, we analyzed the expression profiles of all TFs expressed in our datasets[53]. 127 and 88 TFs were detected at E11 and E10, respectively (Supplementary Figs. 12a, 13a). Hierarchical clustering of the expression profiles identified three distinct clusters of TFs (I, II, and III), with their expression peaking in early, intermediate, and late pseudotime, respectively (Fig. 4a (E11) and Fig. 4b (E10)). Cluster I contained TFs highly

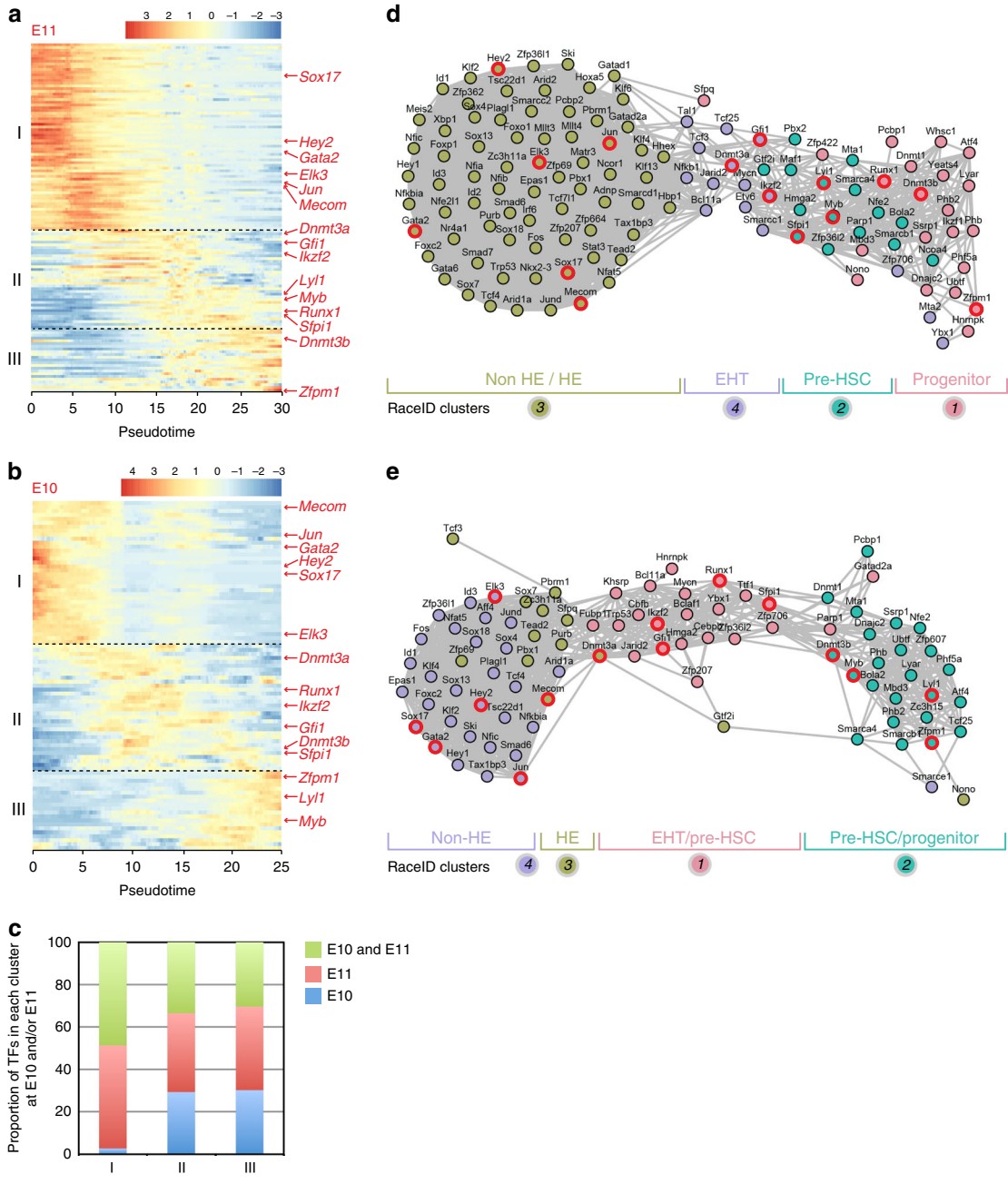

**Fig. 4** Transcription factor networks involved during IAHC formation at E11 and E10. **a, b** Heatmap of differentially expressed transcription factors (TFs) at **a** E11 and **b** E10 between the different cell types (non-HE, HE, EHT, IAHC cells, and YS HSPCs), according to pseudotime. Three distinct expression patterns are identified as clusters I, II, and III. **c** Proportion of transcription factors that are present in clusters I, II, or III at both E10 and E11, only at E11 or only at E10. **d**, **e** Network of TFs related to different expression patterns as determined in **a** and **b**, respectively. Color-coded circles are related to RaceID analysis shown in Figs. 2d and 3c, respectively. TFs mentioned in the results section are outlined in red

expressed in endothelial cells (e.g., Sox17, Hey2)[54,55] (Fig. 4a, b; Supplementary Figs. 12b, c, 13b, c) and TFs upregulated in HE cells (e.g., Elk3, Jun, Mecom)[37]. Gene set enrichment analysis (GSEA) indicates that cluster I TFs were involved in processes such as the regulation of vasculature development, vasculogenesis, or dorsal aorta morphogenesis (Supplementary Figs. 12d, 13d). Cluster II contained TFs expressed in EHT cells and pre-HSCs (Fig. 4a, b). GSEA indicates that these TFs were involved in stem cell differentiation, haematopoiesis, or in the regulation of stem cell maintenance (e.g., Gfi1, Sfpi1, Dnmt3a, Ikzf2)[37,44] (Supplementary Figs. 12e–g, 13e–g). Finally, cluster III contained

TFs that were upregulated in IAHC pre-HSCs and/or progenitors and YS HSPCs (Fig. 4a, b). These TFs were involved in definitive haematopoiesis, negative regulation of cell growth, as for example Zfpm1, which is involved in megakaryocytic and erythroid differentiation (Supplementary Figs. 12h–j, 13h–j). Of note, a large proportion of TFs were common between E10 and E11 in each cluster (48.6% in cluster I, 33.4% in cluster II, and 30.3% in cluster III) (Fig. 4c).

Transcriptional regulatory networks were built based on the Pearson correlations between all TFs at E11 and E10 (Fig. 4d, e, respectively). The TFs upregulated in non-HE and HE cells highly

correlated with each other and with some of the TFs upregulated in EHT cells that still had an active endothelial program. The rest of TFs upregulated in EHT cells correlated with TFs upregulated in IAHC pre-HSCs, and were most likely involved in the initiation of the haematopoietic program. Finally, some TFs upregulated in IAHC pre-HSCs correlated with TFs upregulated in YS HSPCs that exhibited a strong haematopoietic signature. Overall, the TF regulatory networks can be used to explore the TFs involved in the continuous progression of non-HE to HE

cells to EHT cells and finally into IAHC resident pre-HSCs and progenitors.

**Ventral and dorsal IAHCs have similar transcriptomes**. Mouse embryos differ from other vertebrate embryos by the presence of IAHCs in both ventral and dorsal sides of the aorta[10]. To compare the transcriptome of IAHCs according to their position within the aorta (ventral vs. dorsal side), we developed a new

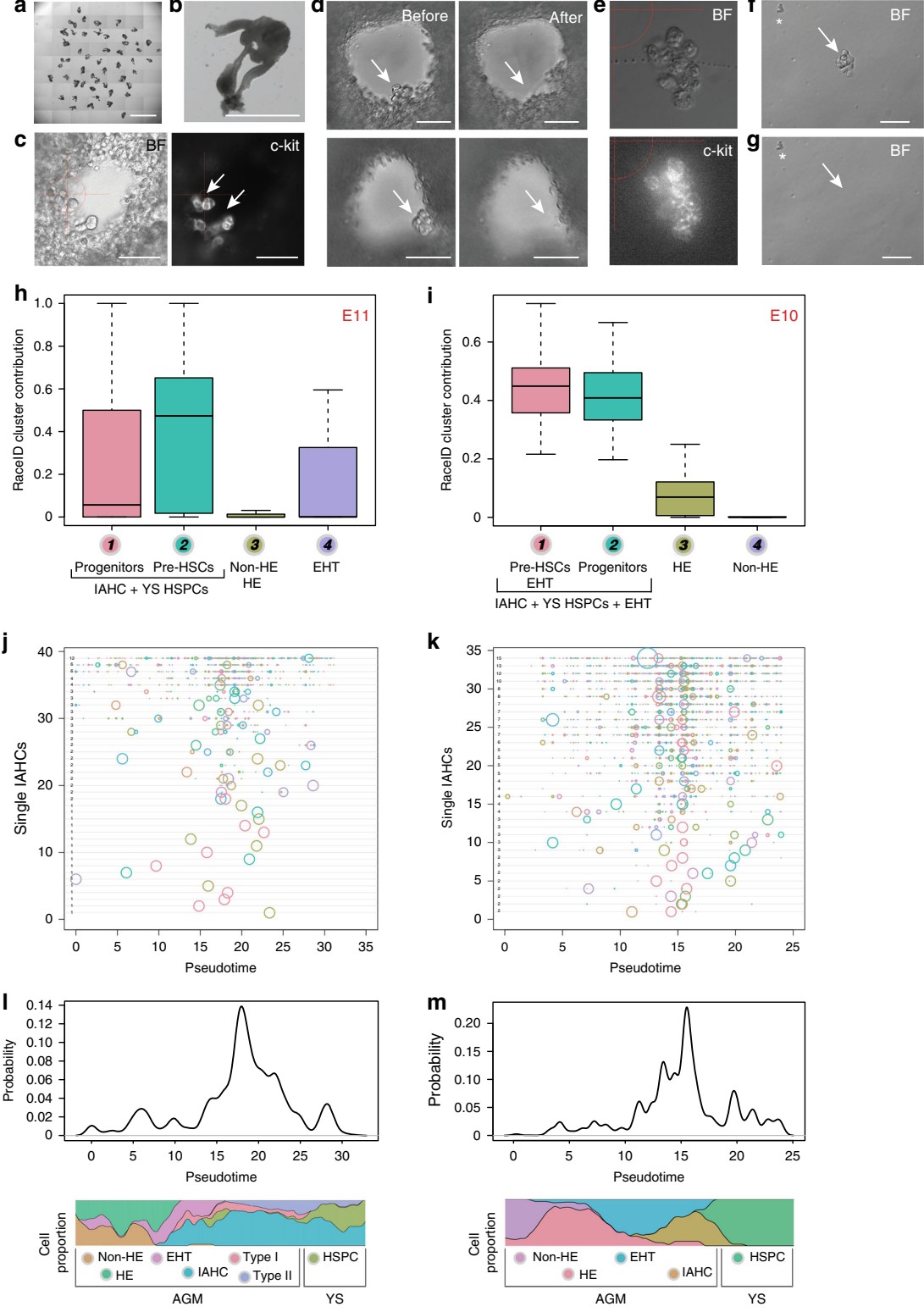

technique to mechanically pick-up single wIAHCs in the aorta of E10 and E11 embryo slices (Fig. 5a, b). An automated cell picker (CellCelector[TM]) (Supplementary Fig. 14a) was used to aspirate single c-kit[+] wIAHCs directly from thick embryo slices (Fig. 5c, d). wIAHCs were then placed in a drop of medium (Fig. 5e, f), picked-up a second time to isolate pure wIAHCs (Fig. 5g) and finally dropped in a tube cap for scRNA-Seq (Supplementary Fig. 14b–f; see Methods for details). To determine the purity of the single wIAHCs and therefore to validate the isolation procedure, the transcriptome of single wIAHCs and single aortic cells (Fig. 1b) were correlated (Supplementary Fig. 14g). The analysis showed that wIAHCs were not contaminated with non-IAHC cells, as it was the case for IAHC cells sorted by flow cytometry. Then, the contribution of each RaceID cluster (Fig. 2d (E11); Fig. 3c (E10)) to each picked wIAHC was calculated. At both E11 and E10, all single wIAHCs were solely composed of IAHC and EHT cells (Fig. 5h, i, respectively). No or negligible endothelial contribution to wIAHCs was observed at E11 and E10. Therefore, our technique allows the isolation of pure single wIAHCs at both E10 and E11, without contamination by non-IAHC cells or endothelial cells.

The transcriptome of E11 and E10 single wIAHCs was then correlated to the transcriptome of E11 and E10 single aortic cells ordered along the pseudotime axis (Fig. 5j–m). At E11, single wIAHCs were projecting mainly between pseudotime points 15 and 25 where IAHC pre-HSCs were present (Fig. 5j, l). At E10, wIAHC projected mainly between pseudotime points 10 and 20 where EHT and IAHC cells were located (Fig. 5k, m). Of note, the number of cells per single wIAHCs (numbers close to the y-axis inside the graph shown in Fig. 5j, k) did not influence their cell composition at both E10 and E11, meaning that larger wIAHCs were not enriched in a specific cell type but were composed of a more heterogeneous group of cells. Overall, wIAHCs are mainly composed of pre-HSCs with less haematopoietic progenitors and EHT cells as embryonic development progresses.

We then compared the transcriptome of ventral and dorsal wIAHCs collected at E10 and E11 (Supplementary Fig. 14f) by clustering on the correlations. Most wIAHCs had similar transcriptome (Fig. 6a). Only three wIAHCs had a low correlation to the rest of wIAHCs (Fig. 6a, bottom left corner of the heatmap). These three wIAHCs had high expression levels of hemoglobin genes, suggesting that they mainly contained erythroid progenitors. DE-Seq analysis was then performed to determine the differentially expressed genes between E11 and E10 ventral and dorsal wIAHCs. Interestingly, only very few genes (out of 18,735 genes) were found differentially expressed (51 genes between E11 and E10 dorsal wIAHCs (Fig. 6b; Supplementary Data 6); 199 genes between E11 and E10 ventral wIAHCs (Fig. 6c; Supplementary Data 7); 2 genes between E11 ventral and dorsal wIAHCs (Fig. 6d; Supplementary Data 8); 16 genes between E10 ventral and dorsal wIAHCs (Fig. 6g; Supplementary Data 9)). In agreement with these data, the contribution of E11 single IAHC cells (Fig. 2d) to E11 ventral and

dorsal wIAHCs (Fig. 6e, f, respectively) was similar, with a majority of pre-HSCs. The contribution of E10 single aortic cells (Fig. 3c) to E10 ventral and dorsal wIAHCs (Fig. 6h, i, respectively) demonstrated that pre-HSCs, progenitors, and EHT cells similarly contributed to both ventral and dorsal E10 wIAHCs. Overall, our analysis revealed that ventral and dorsal wIAHCs are very similar in terms of cellular composition and transcriptome at both E10 and E11. Most transcriptome changes occurred in ventral wIAHCs between E10 and E11, suggesting that these genes might be involved in the pre-HSC maturation process.

## Discussion

The molecular events regulating endothelial specification, IAHC formation, and their consecutive maturation into HSCs remain poorly understood. Using scRNA-Seq, we established that IAHCs contain pre-HSCs and committed progenitors (similar to the ones found in YS). The spatial and temporal comparison of single aortic cells (in silico purified IAHC cells, non-HE and HE cells, cells undergoing EHT) and dorsal/ventral wIAHCs revealed that most transcriptome changes occur between E10 and E11 (in ventral IAHCs), concomitant with EHT and pre-HSC maturation. We provide here an unprecedented resource to study in great details the genes expressed or repressed during endothelial haemogenic specification, EHT, IAHC formation, and pre-HSC maturation toward an HSC fate, as it originally occurs in vivo in the aorta of the mouse embryo.

To determine the transcriptome of IAHC cells, cell isolation to high purity is required. However, isolating an entire IAHC cell population to purity was technically impossible. Therefore, precautions must be taken when analyses are performed on bulk cell measurements, as it does not take into account cell heterogeneity. Consequently, data might reflect the molecular content of contaminating cells (from the mesenchyme or circulating blood) rather than IAHC cells per se. Nevertheless, scRNA-Seq appeared as an excellent tool to in silico purify IAHC cells based on the expression of known IAHC transcripts such as Gpr56[37].

We performed scRNA-Seq to study cell and gene expression heterogeneity in IAHCs at single-cell resolution[56–58], since it has been previously successfully used to examine the transcriptome of rare cell populations[34,59–61]. The comparative transcriptome analysis of in silico purified IAHC cells, sorted AGM pre-HSCs and YS HSPCs, as well as clonogenic experiments, directly addressed the controversial question about the cell composition of IAHCs. Indeed, the discrepancies reported so far about IAHCs were inherent to the different approaches used in different laboratories, the in vitro/in vivo assays being ultimately limited. Moreover, most studies analyzed total AGM or sub-dissected aorta cells but rarely the IAHC population itself. We found that IAHCs contain committed progenitors and pre-HSCs (in accordance to our previous findings[8]), and more surprisingly

**Fig. 5** Transcriptome comparison of pure whole IAHCs mechanically picked-up inside the aorta. **a** E10 embryo slices embedded in agarose. Scale bar, 5 mm. **b** Close-up showing an embryo slice. Scale bar, 1 mm. **c** Close-up showing c-kit[+] IAHCs attached to the aortic endothelium of an E10 embryo slice (left panel, bright filter (BF); right panel, c-kit fluorescence). Scale bar, 50 μm. **d** Two examples (upper and lower panels) of single IAHCs mechanical pick-up. Before pick-up, IAHCs were attached to the ventral side of the aorta (left panel, arrow). After pick-up, IAHCs were no longer visible in the aorta (right panel, arrow). **e** Example of a single mechanically picked-up IAHC (whole IAHC or wIAHC). Upper panel, BF; lower panel, c-kit fluorescence. **f** Presence of a wIAHC (arrow) and contaminating single cells (asterisk) after wIAHC first pick-up. **g** The wIAHC was picked-up a second time to eliminate the contaminating cells that remained in the medium drop (asterisk). **h, i** Box plots showing the RaceID cluster contribution to E11 (**h**) and E10 (**i**) wIAHCs. Color-coded boxes are related to RaceID analysis shown in Fig. 2d and Fig. 3c, respectively. **j, k** Probability for each cell to contribute to single wIAHCs at E11 (**j**) and E10 (**k**). Each horizontal line depicts a single wIAHC. The size of the circle is proportional to the probability for the cell at that position in pseudotime to contribute to that particular wIAHC. The number (next to the y-axis, inside the graph) depicts the total number of cells present in that particular wIAHC. **l, m** Probability density along pseudotime for contributing to wIAHCs

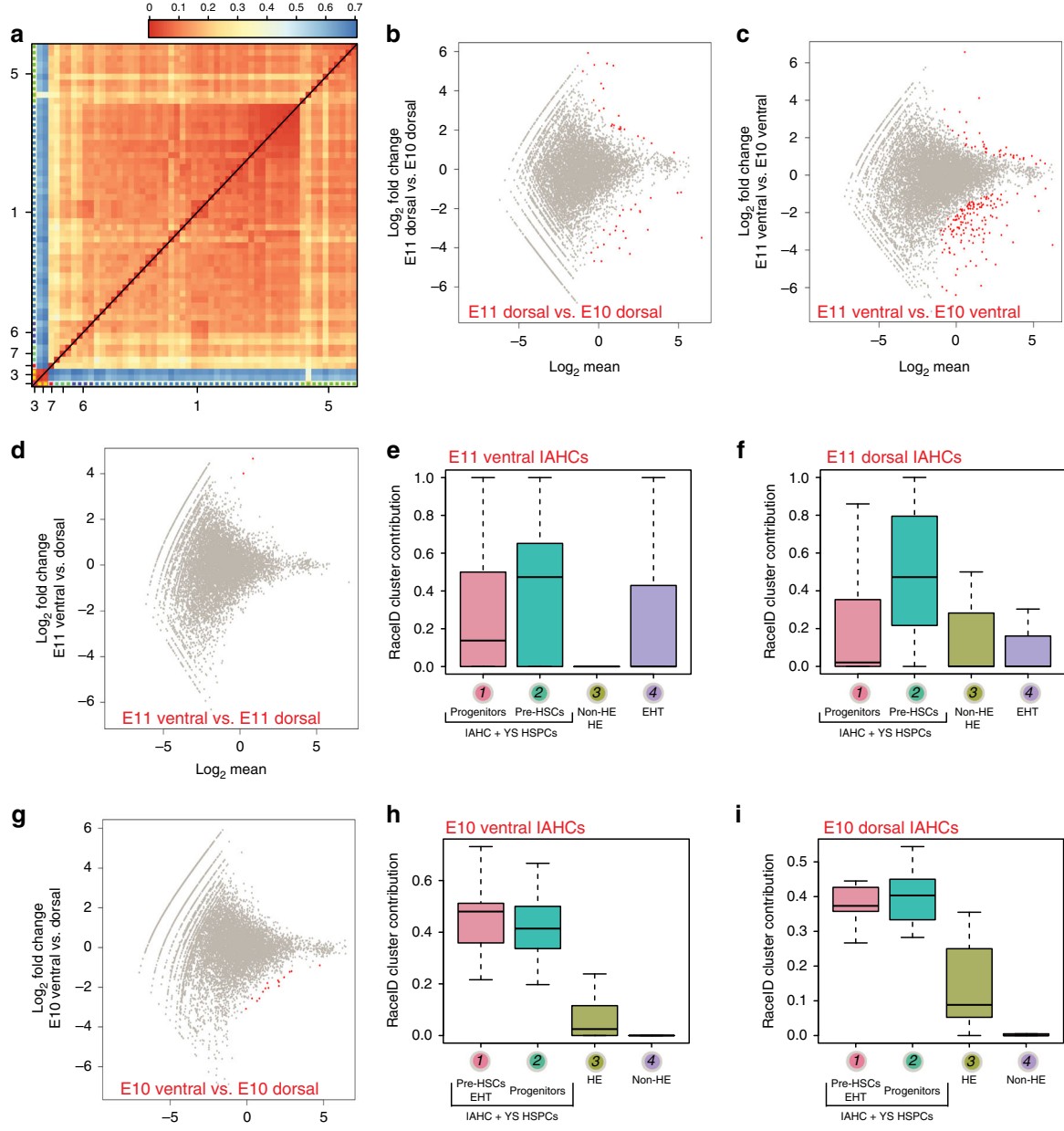

**Fig. 6** Ventral and dorsal wIAHCs have a similar transcriptome at E10 and E11. **a** Heatmap representing the transcriptome similarities measured by the Euclidean distance of the transcriptome correlation matrix for single E10 and E11 wIAHCs. Cluster numbers are shown along the axis for the major clusters. **b–d, g** DE-Seq analysis showing the differential gene expression between E11 and E10 dorsal wIAHCs (51 genes significantly differentially expressed as shown by red dots) (**b**), between E11 and E10 ventral wIAHCs (199 genes) (**c**), between E11 ventral and dorsal wIAHCs (2 genes) (**d**), and between E10 ventral and dorsal wIAHCs (16 genes) (**g**). **e, f** Box plots of the RaceID cluster contribution to E11 ventral (**e**) and dorsal (**f**) wIAHCs. Color-coded boxes are related to RaceID analysis shown in Fig. 2d. **h, i** Box plots of the RaceID cluster contribution to E10 ventral (**h**) and dorsal (**i**) wIAHCs. Color-coded boxes are related to RaceID analysis shown in Fig. 3c

also EMP-like cells. EMPs were described so far only in the YS between E8.5 and E11.5[41,43]. Since EMPs circulate in the blood stream between E10.5 and E12.5 and the EMP-like cells we observed do not express Gpr56 (and do not cluster with committed progenitors), these cells are most likely circulating YS EMPs that attached to IAHCs and were subsequently picked-up in our analysis. Whether the other progenitors (identified in the clonogenic assay and scRNA-Seq analysis) emerged inside the aorta or originated from the YS (as EMPs) and attached to IAHCs while passing through the blood stream remains difficult to ascertain. Others and we have previously demonstrated the

presence of few pre-HSCs in IAHCs (~12–18 pre-HSCs/aorta), capable of long-term multilineage reconstitution upon direct transplantation in neonates[8] or in adult wild type recipients after 6 days of co-culture on stromal cells[29]. However, a large number of type I and type II pre-HSCs, able to produce transplantable HSCs, have been identified in vitro in dissociated/reaggregated culture of E10.5/E11.5 AGMs[30,31,51]. Based on immunostaining observations, IAHCs seem to mostly contain type II pre-HSCs (type I pre-HSCs localizing mainly to the aorta subluminal area)[30,31]. However, scRNA-Seq data revealed that both type I and type II pre-HSCs are present in IAHCs, as

recently reported[47]. Maturing pre-HSCs accumulated at E11, with less type I pre-HSCs at E11 than at E10. Finally, type I and type II pre-HSCs have a very similar transcriptome (only 36 genes differentially expressed out of 18,735 genes). Future research will be necessary to determine whether some of these genes might be involved in pre-HSC maturation and/or the acquisition of transplantation potential.

Transcriptome comparison of wIAHCs, according to their location inside the aorta (ventral vs. dorsal side) at different time points in development, was impossible thus far. The challenge was to access wIAHCs in situ and to detach them from the aortic endothelium without cell damage, wIAHC integrity loss and/or cell contamination (from the surrounding endothelium and mesenchyme tissue). Our mechanical picking technique inside the aorta of non-fixed thick embryo slices allowed the isolation of single and pure wIAHCs. Functional analysis of the sub-dissected aorta initially revealed that HSC activity was restricted to the ventral side of the aorta while clonogenic haematopoietic activity was associated with both ventral and dorsal sides[10]. However, the presence of pre-HSCs and functional HSC activity in both dorsal and ventral side of the aorta was recently reported[62]. Accordingly, we found very few differences in the transcriptome of ventral and dorsal wIAHCs (at both E10 and E11). In agreement with our comparative analysis of single IAHC cells, most transcriptional changes were observed between E10 and E11, mainly in the ventral wIAHCs, which is concomitant with pre-HSC maturation.

More transcriptome changes occurred at E10 (compared to E11) between non-HE, HE, and EHT cells, with more than 500 genes differentially expressed between HE and EHT cells. EHT cells also largely contributed to wIAHCs at E10. One could hypothesize that EHT is particularly active at E10, which would be consistent with the number of IAHC cells peaking at E10.5[9] and the emergence of proliferating type I pre-HSCs at E10.5[32,47]. Further analysis will be needed to determine which genes among those might be involved in the induction of a haemogenic potential in endothelial cells and subsequent EHT. The transcriptional changes, associated with the progressive downregulation of the endothelial program and upregulation of the haematopoietic program, were increased in EHT cells. The endothelial-to-haematopoietic switch, needed for haematopoietic commitment, has been previously reported by performing a functional and transcriptional analysis of single 23GFP+ HE cells (isolated from *Runx1* +23 enhancer–reporter transgenic embryos)[63]. The switch initiated before the detection of HSCs in E9.5 CD41− 23GFP+ HE cells that were still embedded in the endothelial wall, strongly suggesting that haematopoietic specification occurs before EHT. One explanation would be that successive transcriptional switches take place (e.g., in HE cells, EHT cells, pre-HSCs), all being necessary to complete full HSC fate commitment and maturation.

How regulatory programs control cell decision toward an HSC fate remains a central question. The fundamental finding that all HSCs derive from specialized HE cells during embryonic development has paved the way to improve the generation of HSC-like cells in vitro[1,46,64]. More recently, mouse adult endothelial cells[2] and human pluripotent stem cells[3] were converted into transplantable HSCs (via a HE intermediate) by enforced expression of selected TFs and usage of a vascular niche. Interestingly, we found a very similar expression pattern of the TFs used in the above-mentioned studies by looking at our transcriptional regulatory networks and the one from Zhou et al[29]. As expected, Runx1, Spi1, and Gfi1 were expressed in EHT cells and most cluster cells (pre-HSCs and/or progenitors). However, some TFs were either weakly (Hoxa5, Hoxa9, Hoxa10, Fosb) or not (Erg, Lcor) detected. TF requirement might differ in vivo and

in vitro because the culture conditions developed to generate HSCs in vitro do not recreate a complete in vivo microenvironment but instead reproduces few of its elements so far (e.g., addition of cytokines or vascular-niche-derived angiocrine factors, activation/inhibition of specific pathways)[2,3]. Shear stress and 3D cell–cell contact are also obvious regulatory elements that are not (yet) optimally reproduced in vitro. Therefore, some TFs might be needed in vitro but not in vivo due to a redundant role with other TFs in vivo. On the other hand, some factors that need to be discovered are obviously still missing to obtain the generation of HSCs that are fully functional and molecularly similar to bona fide HSCs. Studying gene expression in the physiological context of the aorta is thus essential to understand the in vivo context of HSC production and to lead the way to improve in vitro HSC production.

In conclusion, we provide here a reliable resource of data available for the scientific community to further explore the key factors and pathways involved in the initiation of the HSC program as it occurs physiologically in the aorta in vivo. It is of crucial importance to improve the challenging HSC production in vitro for clinical transplantation purpose.

## Methods

**Embryo generation**. Wild-type mouse embryos were generated from timed matings between C57BL/6 females and males (>8 weeks). Observation of vaginal plugs was considered as embryonic day (E)0 of development. Animals were housed according to institutional guidelines, and procedures were performed in compliance with Standards for Care and Use of Laboratory Animals, with approval from the Dutch Animal Experiment Committee. Wild-type ICR females were mated to Gfi+/tomato/Gfi1b+/GFP males (ICR strain)[44]. All animal work in Manchester was performed under regulations governed by UK Home Office Legislation under the Animals (Scientific Procedures) Act 1986 and was approved by the Animal Welfare and Ethics Review Body (AWERB) of the Cancer Research UK Manchester Institute.

**Embryonic tissue isolation**. Embryos surrounded by the YS were removed from the mother uterus. YS were carefully collected and vitelline vessels were cut off. Embryos were precisely staged according to the day of isolation, the number of somite pairs (sp), and/or based on Thelier criteria. Embryos were collected at embryonic day (E)10 (between 28 and 32 sp) and E11 (>40 sp) of development in PBS supplemented with 10% of Fetal Calf Serum and 5% of Penicillin/Streptomycin (PBS/FCS/PS). The blood was flushed away by injecting PBS/FCS/PS with a glass capillary directly inside the aorta of the embryo trunk, after removal of the head and tail[33].

**Flow cytometry analysis and cell sorting**. IAHC cells were stained by using different procedures (Supplementary Fig. 1). For the TS procedure (Supplementary Fig. 1a), AGM regions were dissected, pooled, and dissociated by collagenase treatment (0.125% collagenase type I in PBS/FCS/PS) for 1 h at 37 °C. After dissociation, cells were washed, stained with PE anti-c-kit (CD117) antibodies (BD Pharmingen, 553355, 1/100), and sorted. For the IAS procedure (Supplementary Fig. 1b, c), embryos were injected with PE anti-c-kit antibodies (and/or AF647 anti-PECAM (CD31) antibodies [BioLegend, 102516, 1/100], CD31 index sorting[65]) directly inside the aorta[12,33]. Non-injected embryos were used as control. After 30 min of staining, AGMs were dissected, pooled, and dissociated by collagenase treatment for 1 h. After dissociation, cells were washed and kept on ice for further analysis. For the c-kit mesenchymal staining (Supplementary Fig. 1e), PE anti-c-kit antibodies were directly injected inside the aorta. AGMs were then dissected and digested, and cells were washed and further stained with FITC anti-CD31 (BD Pharmingen, 553372, 1/100), FITC anti-CD45 (BD Pharmingen, 553080, 1/200), and APC anti-c-kit (BD Pharmingen, 553356, 1/400) antibodies for 30 min. Cells were then washed and kept on ice for further analysis.

For clonogenic assays, PE anti-c-kit and FITC anti-CD31 antibodies were directly injected inside the aorta. AGMs were then dissected and digested, and cells were washed and further stained with APC anti-CD45 (BD Pharmingen, 559864, 1/300) and PE-Cy7 anti-Cdh5 (CD144, BioLegend, 138015, 1/200) antibodies.

For type I and type II pre-HSCs isolation, APC anti-c-kit antibodies were directly injected inside the aorta. AGMs were then dissected and digested, and cells were washed and further stained with FITC anti-CD45 and PE anti-Cdh5 (eBioscience, 12-1441-82, 1/40) antibodies for 30 min at 4 °C (CD45 index sorting).

YS cells were stained after enzymatic digestion with PE anti-c-kit antibodies for 30 min at 4 °C.

For non-HE, HE, and EHT cells, AGMs were dissected, digested, and cells were stained with APC-eFluor780 anti-c-kit (eBioscience, 47-1171-82, 1/200), APC

anti-Cdh5 (eBioscience, 17-1441-82, 1/200), and PerCP Cy5.5 anti-CD45 (BioLegend, 103132, 1/200) antibodies for 30 min and washed[44].

For intra-cytoplasmic staining, cells were stained in surface with APC anti-c-kit and FITC anti-CD31 antibodies for 30 min, washed, fixed, and permeabilised with BD Cytofix/Cytoperm solution kit according to the manufacturer's instructions (Thermo Fisher) and then stained with PE anti-Ikzf2 (BioLegend, 137206, 1/100).

For all sort and flow cytometry analysis, Hoechst or 7AAD was added to exclude dead cells (except for intra-cytoplasmic staining). Cells were analyzed or sorted with a FACSCalibur and an Aria III flow cytometer (BD Biosciences), respectively.

**Mechanical pick-up of single wIAHCs.** Single whole (w)IAHCs were mechanically picked-up by using a CellCelector[TM] (A.L.S.) (Supplementary Fig. 14a). This micromanipulation system combines a motor-operated inverse microscope with a precise robot system to harvest, image, and store parameters and images (e.g., before and after pick-up). The robot platform is equipped with a harvesting module (Supplementary Fig. 14b), usually used to isolate single cells or cell colonies with high precision[66,67]. We adapted the automated robotic system for wIAHC pick-up and manually calibrated the machine by defining parameters for optimal pick-up (e.g., volume and speed for aspiration and for tipping out the harvested material).

The blood was flushed away from the aorta of non-fixed embryo trunk and IAHC cells were stained by intra-aorta injection of anti-c-kit antibodies[12,33]. Embryo trunks were then transversally cut in thick slices of 200 μm with a tissue chopper to visualize c-kit+ IAHC cells inside the aorta (Fig. 5a–c). Slices were transferred into the well of a 6-well plate and immobilized with 0.7% low melting point agarose. When polymerized, the agarose gel was covered with PBS/FCS/PS. The 6-well plate was placed on the motor-operated stage of an inverse Olympus microscope connected to a high-definition CCD Olympus camera and to the robotic platform of the CellCelector[TM]. All embryo slices were manually screened for the presence of c-kit+ IAHCs (Fig. 5c). Bright-field and fluorescence images of embryo slices were taken before and after each wIAHC pick-up (two before/after pick-up examples are shown in Fig. 5d). All pictures and picture information were transmitted to the imaging software and stored. wIAHCs were mechanically harvested one by one, by gentle aspiration with a calibrated glass capillary (Micropipets, Origio Inc.) containing PBS and connected to a syringe placed on a robotic arm (SingleCell-modul) (Supplementary Fig. 14b, c). The aspirated IAHC was then automatically transported to an empty well of a 6-well plate (Supplementary Fig. 14d, left red arrow; Fig. 5e). The deposit drop was screened for pick-up control (e.g., wIAHC integrity, no apparent contaminating cells attached). To avoid any contamination of the collected sample (c-kit+ wIAHC; Fig. 5f, arrow) with c-kit− cells that might detach from the embryo slice during aspiration (Fig. 5f, asterisk), a second pick-up was performed (Supplementary Fig. 14d, right red arrow; Fig. 5g, arrow). For this step, single wIAHCs were transferred directly in the cap of a 0.2 ml tube (Supplementary Fig. 14e) that was immediately closed, spun down and stored at −80 °C. wIAHCs were collected at E10 and E11 in the ventral and dorsal side of the aorta for sc-RNA-Seq (Supplementary Fig. 14f).

**RNA extraction from single IAHC cells and single wIAHCs.** Single cells were sorted in the wells of 96-well PCR plates, each well containing 100 μl of TRIzol reagent and 0.2 μl of a 1:250,000 dilution of ERCC spike ins (Thermo Fisher). wIAHCs were collected after pick-up in tubes containing the above solution. Total RNA was extracted from single cells, according to TRIzol manufacturer's instructions (Thermo Fisher) with few alterations. GlycoBlue reagent (0.2 μl) was added to each sample to facilitate RNA precipitation. After overnight isopropanol precipitation, RNA pellets were air dried for up to 15 min, and suspended in CEL-Seq first-strand primer solution.

**CEL-Seq library preparation.** Single wIAHCs and single cells were processed according to the CEL-Seq protocol[34]. Briefly, single cells (or wIAHCs) RNA obtained after TRIzol extraction was suspended in primer solution, denatured at 70 °C for 2 min and quickly chilled. First-strand synthesis was carried out at 42 °C for 1 h followed by a second-strand synthesis step at 16 °C for 1 h. cDNA was cleaned-up and in vitro transcription was set-up overnight at 37 °C. Amplified RNA was fragmented and cleaned-up for 3′ adapter ligation, reverse transcription reaction, and PCR amplification. Libraries were sequenced with an Illumina NextSeq 2500 by using 75 bp paired end sequencing.

**CEL-Seq data analysis.** Initial analysis was performed using the RaceID2 algorithm, which is a part of StemID[68]. On average, 12,000 unique transcripts were detected per cell. The sequenced cells were down sampled to 5000 unique transcripts. All other cells were discarded from the analysis. Monocle analysis[45] was performed using default parameters. For the dimensionality reduction of Monocle, all genes expressing in at least one cell with at least five transcripts were included. For the differentially expressed genes along pseudotime, a sliding window of 10 Pseudotime arbitrary units was used. The first five units in this window were compared to the second five units to assess the number of differentially expressed genes within the window. For the TF network analysis, the Fantom5 mouse TFs was used. To cope with drop-out effects due to the low expression of most TFs, we included all TFs that were expressed in at least one cell with at least one transcript after smoothening of the data along Pseudotime. GSEA analyses were performed using EnrichR[69,70], with default parameters.

**Comparison of single IAHC cells and wIAHCs.** Comparison of single IAHC cells to wIAHCs was performed by using a Markov–Chain Monte Carlo (MCMC) approach. For each wIAHC, a random number of cells corresponding to the number of cells present in that particular wIAHC was chosen from the single IAHC cells (artificial wIAHC). The collective transcriptome of this artificial wIAHC was then compared to the transcriptome of the wIAHC based on Pearson correlation. Then, 10,000 random swaps of cells within the artificial wIAHC were performed to obtain the set of single cells that had the highest correlation to that particular wIAHC. This entire process was repeated 100 times using different starting cells to ascertain that the starting population of cells did not influence the outcome of the MCMC.

**Single-molecule FISH on embryo cryosections.** Probe libraries were designed (for Gpr56, CD34, Kit, Epha7, Mpo, and Vwf) and fluorescently labeled with Cy5, Alexa Fluor 594, or TMR fluorophores. All probe libraries consisted of 20–39 oligonucleotides of 20 bp lengths complementary to the coding sequence of the genes of interest. Cryosections (8 μm) were obtained and mounted on poly-l-lysine-coated cover glass. Cryosections were then fixed in 4% paraformaldehyde (PFA) for 15 min and dehydrated in 70% ethanol for a minimum of 2 h. Embryo cryosections were washed and hybridized overnight with selected probes at 30 °C. DAPI staining was done after washes to visualize nuclei. Images were acquired on a Perkin Elmer spinning disc confocal microscope with a 100× oil-immersion objective. Images were processed and combined using ImageJ.

**Immunostaining on live thick embryo slices and cryosections.** Prom1 (CD133) staining was performed on live thick E11 embryo slices[12,33]. Briefly, the blood was flushed out of the aorta and FITC anti-CD31, PE anti-CD133 (BioLegend, 141203, 1/100), and APC anti-c-kit antibodies were injected directly inside the aorta. After 30 min of incubation, the embryos were transversally cut in thick slices, embedded in agarose gel and imaged with a Zeiss LSM700 confocal microscope.

Gja5 (Connexin 40) staining was performed on embryo cryosections (10 μm). Briefly, E10 embryos were fixed (4% PFA), cryoprotected in 15% sucrose buffer and frozen in liquid nitrogen. Sections were collected on Superfrost slides (Thermo), rehydrated in PBS, blocked with PBS/1% BSA/0.1% Triton-X100 for 1 h. Sections were incubated overnight at 4 °C with primary antibodies (purified anti-Connexin 40 (Gja5, Thermo Fisher, 37-8900, 1/250) and purified anti-Runx (Abcam, Ab92336, 1/250)). After several washes in PBS, sections were incubated for 90 min at room temperature with secondary antibodies (Alexa 647 Goat anti-mouse IgG1 [Thermo Fisher, A21240, 1/250] to reveal Connexin 40 and Alexa 488 Goat anti-rabbit [Thermo Fisher, A11008, 1/250] to reveal Runx). After several washes in PBS, sections were incubated for 1 h at room temperature with PE anti-CD31 antibody. After wash, sections were stained with DAPI to visualize nuclei.

**In vitro clonogenic assay.** AGM cells isolated from E11 embryos and sorted based on c-kit, CD31, CD45, and Cdh5 differential expression were plated in methylcellulose (M3434; StemCell Technologies). After 12 days of culture at 37 °C, colonies were counted after microscopic observation. Five types of clonogenic progenitors were identified: CFU-GEMM (Colony Forming Unit-Granulocytes Erythrocytes Macrophages Megakaryocytes), CFU-GM (CFU-Granulocytes Macrophages), CFU-M (CFU-Macrophages), CFU-G and BFU-E (Burst Forming Unit-Erythroid).

**Code availability.** The transcriptome analysis was performed using RaceID2, available at https://github.com/dgrun.

**Data availability.** The datasets generated during and/or analyzed during the current study are available in the Gene Expression Omnibus repository, with the series record GSE112642.

The authors declare that all data supporting the findings of this study are available within the manuscript or its Supplementary Files or are available from the corresponding author upon reasonable request.

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

## Acknowledgements

The authors thank the lab members for helpful discussion and various technical help. We would like to thank Jens Eberhardt and Michael Hollstein for their help to develop the automated cell picking of wIAHCs with the CellCellector™. We also thank the Animal Facility for mouse care and the Optical Imaging Center for confocal microscope access (both at the Hubrecht Institute). We thank Reinier van der Linden (Hubrecht Institute), and Jeroen van Velzen and Pien van der Burght (UMC Utrecht) for help with cell sorting. This work was supported in CR lab by a European Research Council grant (ERC, project number 220-H75001EU/HSCOrigin-309361), a TOP8861-subsidy from NWO/ZonMw (912.15.017), and the UMC Utrecht "Regenerative Medicine & Stem Cells" priority research program, in G.L. and V.K. labs by the Medical Research Council (MR/P000673/1), the Biotechnology and Biological Sciences Research Council (BB/I001794/1), Bloodwise (12037), the European Union's Horizon 2020 (GA6586250) and Cancer Research UK (C5759/A20971), and in A. v O. lab by a TOP-subsidy NWO/CW (714.016.001).

## Author contributions

C.S.B. and L.K. are co-first authors, and A.K. and J.-C.B. are co-second authors. C.R. conceived ideas and designed the research; C.R. performed embryo dissections with the help of C.S.B. and A.K.; C.S.B. performed AGM IAHC cells and YS cells staining and sorting, and wIAHC mechanical pick-up. V.K. and G.L. contributed to the design of sorts for non-HE, HE, and EHT cells. R.T. performed AGM non-HE, HE, and EHT cell staining and sorting. L.K. and C.S.B. performed scRNA-Seq. C.S.B. performed smFISH experiments with the help of L.K.; L.Y. performed immunostaining and flow cytometry experiments; L.K. analyzed all scRNA-Seq data with the help of A.vO., C.R., C.S.B., and J.-C.B.; A.K. performed cell sorting and in vitro clonogenic assays. J.-C.B. contributed to initial experimental design and experiments. C.R. analyzed and interpreted the experiments with the help of the other authors. C.R. created the figures and wrote the paper. All authors commented on the manuscript.

## Additional information

**Competing interests:** The authors declare no competing interests.

