## [Peer Review File · Nature Communications]

Reviewers' comments:

Reviewer #1 (Remarks to the Author):

The authors performed single cell RNA-seq using IAHCs of the E10 and E11 AGM to understand molecular events regulating hemogenic endothelial cell specification, IAHC formation and their maturation into HSCs. Particularly, nonHE, HE, pre-HSC, and EHT cells were isolated and subjected to single cell RNA seq. The data suggests that IAHCs contain both preHSCs and hematopoietic progenitors and that as cells mature into hematopoietic cells, endothelial program becomes down regulated while hematopoietic program becomes activated. This work makes a good usage of Gfi1 and Gfi1b reporters to isolate various cell populations presumably representing EC, HE, EHT, preHSC, and hematopoietic progenitors. This work also reports the method to isolate IAHC cells. Overall, this manuscript provides detailed molecular analysis of the dynamic transcriptional changes occurring in the AGM. There are some issues however where I feel the paper could be improved, as described below.

This study is currently limited to the analysis of the transcriptome and molecular profiling. There is no functional validation whether the genes that show transcriptional changes in different cell populations could indeed have role in that particular cell population.

There is a gap between HE/EHT cells and IAHCs in the t-SNE plot of the single cells in Figure 2d and Figure 3c. Also, in Figure 2e, Cdh5, Kit, and Ptprc gene expression dynamics are not continuous in pseudotime. The authors sorted Gfi1b⁻ cells from Cdh5⁺ cells as HE or EHT cells. Could this sorting strategy cause some cKIT⁺ cells to be missed? Alternatively, the authors also mentioned that IAHCs undergo extensive cell proliferation. Is it possible if the cell cycle status could affect the analysis of the gene expression pattern of the EHT cell population?

Figures 2 and 3 display the progression of the IAHCs from preHSCs to progenitors. To that end, it is recommended to in silico analyze pre-HSCs, type I and type II, in relationship to CD41^{low} and CD201^{high} preHSCs, that were characterized in the Zhou paper (Nature 2016). Also, it would be informative if the authors compare their data to that of Zhou's (Nature, 2016) including fetal liver HSCs from E12 and E14.

Figure 4 shows transcriptional changes of the transcription factors in the AGM cell populations. It is striking to see that transcription factors expressed barely overlap between E10 and E11 HE or EHT cells. It is hard to believe that EHT cells of E10 and E11 AGM would be that different in terms of transcription factor transcriptome. It would be great to see some validations whether indeed temporal changes in transcription factor genes are necessary for the AGM hematopoiesis. It is intriguing that Gata2 expression is specifically enriched in nonHE, not in preHSC or EHT, in both E10 and E11, given the literature on Gata2 in HSCs.

Supplementary Figure 7 shows overlap between upregulated genes in IAHCs from RaceID cluster 1 and BM progenitors, and upregulated genes in IAHC cells from RaceID cluster 2 and BM HSCs. It would be informative if the analysis be done using the whole transcriptome, not upregulated, to that of BM and FL progenitors and HSCs.

Currently, no signaling pathways are described in the transition from nonHE, HE, pre-HSC, and EHT. It would be good to see an analysis and discussion of the previously described signaling pathways reported functioning in the AGM hematopoiesis (for example, Notch, BMP, and cKIT) in relationship to some of the transcription factors such as Gata2. Additionally, Zhou et al. identified mTORC2 signaling to be highly enriched in pre-HSC-I compared to ECs (Nature, 2017). Perhaps, the authors can in silico analyze mTORC2-signaling pathway in their cell populations.

Sugimura et al. recently used ERG, HOXA5, HOXA9, HOXA10, LCOR, RUNX1 and SPI1 to generate HSPCs from human PS cells (Nature, 2017). Lis et al. used Fosb, Gfi1, Runx1, and Spi1 to

reprogram endothelial cells to functional HSCs (Nature, 2017). Intriguingly, these factors seem to be expressed in a nondiscriminatory manner in nonHE, HE, pre-HSC, and progenitors. Thus, it is not so intuitive how the data from this manuscript can be useful for the future HSC generation or reprogramming.

Reviewer #2 (Remarks to the Author):

Summary:

The molecular mechanisms that drive the emergence of HSCs from hemogenic endothelium has been to dissect because of the difficulty in profiling single cells. More generally, the spectrum of cellular heterogeneity as HSCs are specified and as they emerge has likewise been difficult to characterize for the same reason. In this manuscript Baron et al use scRNA-Seq to address these questions, and couple this with some smFISH validation, and they showcase a automated cell picking technology to compare dorsal versus ventral IAHCs transcriptomes. Overall, this is a solid contribution, and I have only minor suggestions as listed below:

- The tSNE plots do not agree with the RACE ID cluster assignments, and therefore the tSNE plots themselves, while popular, are somewhat confusion and perhaps not the most appropriate way to present the scRNA-Seq data.
- Figure 1: The authors show scRNA-Seq of Kit, but what about the concordance between marker expression and RNA for the others (CD45 and CD31)?
- in silico purification of IAHC: what threshold used to define Gpr56+? Some non-IAHC cells appear to be positive in Figure 1.
- Pg4: "scRNA-seq represents a powerful tool to correct for cell contamination by in silico purifying IAHC cells based on the expression of known IAHC marker genes (such as Gpr56" <-- This claim depends on the sensitivity of the scRNA-Seq. If there are libraries in which Gpr56 has 'dropped out', then this method will fasley assign the cell to a non-IAHC group. What is the CEL-Seq sensitivity? Can this issue of drop out be addressed to some extent by using drop out correction methods such as MAGIC?
- RACE ID cluster labes in Figure 2D are not clear. There are several points that have IDs like 13, 14 etc. Are these just cells in close proximity by different RACE ID clusters? And there are even cells labeled as cluster 7, but the llegend only goes up to 4. Related to this, how are 'main' clusters identified (Pg6)? Is this just a call made by eye?
- Figure 3A: The EHT cluster appears to be detected as a separate branch by Monocle. Can the authors explain this?
- What units are the mean columns in the supp tables? The values are quite large
- Are there any hints of genes/signaling pathways related to dorsal ventral patterning in the D vs V comparisons?
- The authors claim "It will pave the way for improving HSC production in vitro". But the authors do not discuss any of the new transcription factors that emerge from their pseudotime analysis. What new knowledge has been gained here? Performing an GSEA-style analysis of signaling pathways that correlate with the pseudotime analysis would, in principle, be helpful to propose new means to coax iPSC-HE to HSCs.

Response to the Reviewer 1 comments

The authors performed single cell RNA-seq using IAHCs of the E10 and E11 AGM to understand molecular events regulating hemogenic endothelial cell specification, IAHC formation and their maturation into HSCs. Particularly, nonHE, HE, pre-HSC, and EHT cells were isolated and subjected to single cell RNA seq. The data suggests that IAHCs contain both preHSCs and hematopoietic progenitors and that as cells mature into hematopoietic cells, endothelial program becomes down regulated while hematopoietic program becomes activated. This work makes a good usage of Gfi1 and Gfi1b reporters to isolate various cell populations presumably representing EC, HE, EHT, preHSC, and hematopoietic progenitors. This work also reports the method to isolate IAHC cells. Overall, this manuscript provides detailed molecular analysis of the dynamic transcriptional changes occurring in the AGM. There are some issues however where I feel the paper could be improved, as described below.

→ We thank Reviewer 1 for his/her appreciation of our data and manuscript, and also for his/her questions and constructive comments that we answered below. According to Reviewer 1 requests, we performed new data analyses (see below Supplementary data). We incorporated the most relevant data and findings in the revised version to improve the manuscript. We are of course open to add other supplementary data if Reviewer 1 believes that it is necessary.

- This study is currently limited to the analysis of the transcriptome and molecular profiling. There is no functional validation whether the genes that show transcriptional changes in different cell populations could indeed have role in that particular cell population.

→ The goal of the study is to provide the first detailed and precise analysis of the transcriptome and molecular profiles of the successive populations that progressively commit towards a HSC fate. It was done for the first time according to both the developmental time point (E10 and E11) and the spatial organization of IAHCs inside the aorta (dorsal or ventral side), by analyzing more than 500 single cells. We agree with Reviewer 1 that the functional validation of the genes found differentially expressed in our study is of great interest and will be the logical follow-up research of the present study. However, we believe that this time-consuming research is out of scope of the present study.

However, to reinforce the interest of the genes highlighted in our study, we performed different immunostaining techniques on AGM cells from E10 or E11 embryos for a selection of three new markers. They were chosen for their differential expression in our populations of interest and for the commercial availability of antibodies. These new data are presented in **Supplementary data 1** (see below) and incorporated in the revised manuscript as new **Supplementary Fig. 10**. We found, as predicted by our sc-RNA-seq data analysis, that:

- Ikzf2 (Helios) is much highly expressed at the protein level in c-kit⁺CD31⁺ IAHC cells (86.6%) compared to endothelial cells (15.8%) (**Supplementary data 1a,b**), as shown by flow cytometry analysis of E11 AGM cells.

- Prom1 (CD133) is expressed by c-kit⁻CD31⁺ endothelial cells but not by c-kit⁺CD31⁺ IAHC cells (**Supplementary data 1c-g**), as shown by immunostaining on live thick E11 embryo slices.

- Gja5 (Connexin 40) is expressed by CD31⁺ endothelial cells but not by Runx1⁺CD31⁺ IAHC cells (**Supplementary data 1h-l**), as shown by immunostaining on E10 embryo cryosections. We believe that the good correlation between our sc-RNA-seq data and protein expression are very promising. It will lead the way to study the functionality of specific genes of interest in specific populations of the aorta or to test their usefulness as new surface markers for specific cell isolation in the future.

We have added the following paragraph in the revised manuscript: “*To reinforce the relevance of the genes highlighted in our study, we performed different immunostaining*

techniques on AGM cells from E10 or E11 embryos for a selection of three new markers. *IKZF2* (Helios), *PROM1* (CD133) and *GJA5* (Connexin 40) were chosen for their differential expression in our populations of interest (endothelium versus IAHC cells) and for the commercial availability of antibodies. As predicted by our scRNA-seq data, *IKZF2* was much highly expressed at the protein level in *c-kit*⁺*CD31*⁺ IAHC cells (86.6%) than in endothelial cells (15.8%), as shown by flow cytometry analysis (**Supplementary Fig. 10 a,b**). On the other hand, *PROM1* and *GJA5* were expressed by *CD31*⁺ endothelial cells but not by IAHC cells (*c-kit*⁺*CD31*⁺ or *Runx1*⁺*CD31*⁺ cells, **Supplementary Fig. 10c-g** [immunostaining on live E11 thick embryo slices] and **Supplementary Fig. 10h-l** [immunostaining on E10 embryo cryosections], respectively). There is a good correlation between scRNA-seq data and protein expression, which consolidates our datasets. Such datasets will be useful to study the functionality of specific genes of interest in specific populations of the aorta or to test their usefulness as new surface markers for specific cell isolation in the future.”

- There is a gap between HE/EHT cells and IAHCs in the t-SNE plot of the single cells in Figure 2d and Figure 3c. Also, in Figure 2e, *Cdh5*, *Kit*, and *Ptprc* gene expression dynamics are not continuous in pseudotime. The authors sorted *Gfi1b*⁻ cells from *Cdh5*⁺ cells as HE or EHT cells. Could this sorting strategy cause some *cKIT*⁺ cells to be missed?

→ We agree with Reviewer 1 that there is a gap between HE/EHT cells and IAHC cells on the t-SNE maps at E10 and E11. This gap could reflect a batch effect because the amount of each population that can be collected at E10 and E11 per embryo is very low. It therefore inevitably involves that the experiments were performed in different batches (e.g. cells collected from different litters, on different dates of experiments, and on different sorters (due to logistic reasons)). Nevertheless, we found only few batch effect related genes such as L or S ribosomal proteins (*Rpl*, *Rps*) or stress related genes (among a list of 1,845 differentially expressed genes) by comparing the genes differentially expressed in the two cell populations that appear separated by the gap (as represented in red and blue in **Supplementary data 2a**). The use of Principal Component Analysis (PCA) to visualize the data reduced the gap (**Supplementary data 2b**), indicating that the gap observed on the t-SNE representation most likely reflects the imperfection of the t-SNE algorithm rather than a lack of continuity between the different populations. Moreover, we sorted all *c-kit*⁺ cells from the AGM (IAHC and non-IAHC cells) so we believe that no *c-kit*⁺ cells were missing in our analysis.

→ The question of Reviewer 1 most likely referred to Fig.2f (and not Fig.2e, which is a DE-Seq plot of type II versus type I pre-HSCs). Of note, Fig.2f does not show gene expression but the fluorescence intensity (which reflects protein expression for *CDH5*, *c-kit*, and *PTPRC*) that was recorded during the index sorting of type I and type II pre-HSCs. We agree with Reviewer 1 that there are some fluctuations in the fluorescence intensity along pseudotime. Pseudotime ordering was done based on gene expression and not protein levels. Although on a large pseudotime scale protein expression nicely coincides with gene expression levels (e.g. *PTPRC* protein levels increased as pseudotime progresses), some imperfections can be explained by small discrepancies between RNA and protein expression (e.g. minor fluctuations of *PTPRC* protein levels between pseudotime 10 and 15). The fluorescence intensities were shown in pre-HSCs. Therefore, the diagram is restricted between pseudotime points 10 and 22, where pre-HSCs are found in IAHCs.

Alternatively, the authors also mentioned that IAHCs undergo extensive cell proliferation. Is it possible if the cell cycle status could affect the analysis of the gene expression pattern of the EHT cell population?

→ We thank Reviewer 1 for this important question. We indeed mentioned in the discussion that IAHC cells proliferate, as recently reported in Batsivari, A. *et al.* (2017) and

Rybtsov, S. *et al.* (2016). To test whether the cell cycle status could affect the analysis of the gene expression pattern of our cells, we examined the pathway “cell cycle” according to pseudotime by using the KEGG database. As shown on the heatmap (**Supplementary data 3**), many genes of the cell cycle pathway were detected in our E11 dataset. Most of them were expressed in most cell populations (ordered along the pseudotime axis on the heatmap). Using the new version of RaceID (RaceID3, manuscript under review, https://github.com/dgrun/RaceID3_StemID2), we then clustered our data after correction for cell cycle and proliferation genes. As shown in **Supplementary data 4** and **Supplementary data 5**, we observed that while the shape of the t-SNE map slightly changed, gene expression did not change in the different populations compared to our initial analyses (Supplementary Fig.4 and Supplementary Fig.6). Of note, no gap was visible between EHT and IAHC cells. It therefore supports our answer above that no populations are missing in our analysis.

We included the heatmap in the revised version of the manuscript as new **Supplementary Fig. 5** but not the reanalysis because (i) the proliferation status of IAHC cells was just mentioned in the discussion in regard to the recent literature but was not the focus of our analysis and (ii) the above mentioned analyses revealed that the cell cycle and proliferation state of the cells does not affect the gene expression pattern analysis and thus our conclusions. We have added the following sentence in the revised manuscript: “*The proliferation of IAHC cells was recently reported^{35, 51}. Using the KEGG database, we analyzed the “cell cycle” pathway according to pseudotime and indeed detected many genes of the cell cycle pathway in most cell populations of our E11 dataset (Supplementary Fig. 5). Importantly, the gene expression observed in the different populations (as reported above) did not change after correction for cell cycle and proliferation genes.*”

- *Figures 2 and 3 display the progression of the IAHCs from preHSCs to progenitors. To that end, it is recommended to in silico analyze pre-HSCs, type I and type II, in relationship to CD41^{low} and CD201^{high} preHSCs, that were characterized in the Zhou paper (Nature 2016).*

→ For our analysis we sorted pre-HSCs type I and type II based on surface marker expression, as previously reported (c-kit⁺Cdh5⁺CD45⁻ and c-kit⁺Cdh5⁺CD45⁺, respectively) (Taoudi *et al.* 2008, Rybtsov *et al.* 2011). *In silico* analyses of these cells in relationship to CD41^{low} and CD201^{high} expression is delicate. The expression level of CD41 (*Itga2b*) is very low and it might result in discarding pre-HSCs that appear CD41⁻ in our dataset at E11 and E10 (**Supplementary data 6a and b**, respectively). Moreover, our data clearly show that CD201 (*Procr*) is not only expressed by pre-HSCs but also by EHT and HE cells at E11 and E10 (**Supplementary data 6c and d**, respectively). Therefore, *in silico* purifying pre-HSCs based on CD41 and CD201 expression might result in discarding part of the pre-HSC population and including part of EHT cells. In addition, the comparative analysis requested below between our populations and the ones from Zhou *et al.* (2016) shows that our pre-HSCs type I and type II populations are comparable to the Zhou *et al.* populations, regardless of the surface markers used for cell sorting strategy (**Supplementary data 7**).

- *Also, it would be informative if the authors compare their data to that of Zhou’s (Nature, 2016) including fetal liver HSCs from E12 and E14.*

→ As requested by Reviewer 1, we compared our E11 pre-HSC populations type I and type II (individually) to the populations described in the publication of Zhou *et al.* (Nature, 2016). Zhou *et al.* analyzed 4 populations in E11 AGM: (1) t1 CD201^{low} (CD31⁺CD45⁻CD41^{low}c-kit⁺CD201^{low}), (2) t1 CD201^{high} (CD31⁺CD45⁻CD41^{low}c-kit⁺CD201^{high}), (3) t2 CD41^{low} (CD31⁺CD45⁺CD41^{low}) and (4) t2 CD201^{high} (CD31⁺CD45⁺c-kit⁺CD201^{high}). They also isolated E12 fetal liver cells (Lin⁻Sca1⁺Mac1^{low}CD201⁺) and adult bone marrow HSCs. We first established lists of genes that were differentially expressed in each E11 AGM population versus the three other populations (based on Zhou paper data);

second, we established lists of genes that were differentially expressed in each of our pre-HSC populations (type I or type II) versus our other populations (non-HE, HE, EHT and IAHC cells, based on the present manuscript data); third, we compared the lists from the Zhou paper and our lists of genes expressed by the pre-HSC populations; finally, we obtained a p-value representing the significance of the overlap between the populations from the Zhou paper and our pre-HSC populations. Results are shown in **Supplementary data 7**.

This analysis shows that:

- our pre-HSC type I population had a significant high overlap with the t1 CD201^{high} population (shown to be enriched in pre-HSCs type I) (**Supplementary data 7a [left panel] and 7b** (42% of genes in common)). On the other hand, our pre-HSCs type I had no significant overlap with the t2 CD41^{low} population (shown to be enriched in pre-HSCs type II) (**Supplementary data 7a [left panel] and 7c** (7% of genes in common)).

- Our pre-HSC type II population had also a significant high overlap with the t1 CD201^{high} population, although lower than the pre-HSC type I population (**Supplementary data 7a [right panel] and 7d** (27% of genes in common)). Also, our pre-HSCs type II had a significant high overlap with the t2 CD41^{low} population (shown to be enriched in pre-HSCs type II) (**Supplementary data 7a [right panel] and 7e** (21% of genes in common)).

Accordingly, the t1 CD201^{high} population (similar to our pre-HSC type I cells) expressed *Gpr56*, *Pecam1* and *Kit* but not yet *Ptprc* (**Supplementary data 7f**). The t2 CD41^{low} population (similar to our pre-HSC type II cells) expressed *Gpr56*, *Pecam1*, *Kit* and *Ptprc* (**Supplementary data 7f**).

The t2 CD41^{low} population was shown to be contaminated by committed progenitors in Zhou *et al.*. Accordingly, we found that our progenitor cell population has a significant high overlap with the t2 CD41^{low} population (**Supplementary data 7g and 7h** (21% of genes in common)).

Our pre-HSC populations (type I and II) did not overlap with the t1 CD201^{low} population (shown to have no transplantation potential) (**Supplementary data 7a**). However, the t1 CD201^{low} population had a very high overlap with our non-IAHC cell population (**Supplementary data 7i and 7j** (28% of genes in common)). Accordingly, the t1 CD201^{low} population did not express *Gpr56* and *Ptprc* (**Supplementary data 7f**).

As expected, both pre-HSC populations had a significant overlap with the AGM t2 CD201^{high} cells, E12 FL HSCs (**Supplementary data 7a**) and adult BM HSCs (**Supplementary data 7a, 7k** (27% of genes in common) **and 7l** (18% of genes in common)).

Overall, this comparative data analysis confirmed the committed progenitor identity of RaceID cluster 1 cells and the pre-HSC identity of RaceID cluster 2 cells (Fig. 2d). It also shows that our datasets provide complementary information and the possibility to perform comparative analyses with previously published datasets (Zhou *et al.* [in the revised version], Lara-Astiaso *et al.* [in the original version of the manuscript]), independently of the markers used for cell sorting strategy.

The initial comparative analysis of our data with the one from Lara-Astiaso *et al.* (Science 2014) has been removed and replaced by the above comparative data analysis with the Zhou *et al.* paper (Nature 2016). The analysis and findings are included in the text of the revised manuscript (as described above) and most relevant data (as reported below in **Supplementary data 7**) are depicted in the new **Supplementary Fig. 8**.

- *Figure 4 shows transcriptional changes of the transcription factors in the AGM cell populations. It is striking to see that transcription factors expressed barely overlap between E10 and E11 HE or EHT cells. It is hard to believe that EHT cells of E10 and E11 AGM would be that different in terms of transcription factor transcriptome.*

→ In fact, a large proportion of transcription factors (TFs) are common between E10 and E11 in each cluster (48.6% in Cluster I, 33.4% in Cluster II and 30.3% in Cluster III).

Reviewer 1 might have thought that only the TFs circled in red in the original Fig.4b and 4d were in common between E10 and E11. It is not the case. They correspond to the TFs mentioned on the side of the heatmaps (original Fig.4a and 4c). We apologize for the confusion. It is an important point and we added a graph, representing the proportion of TFs that are present at both E10 and E11, only at E11 or only at E10 in Clusters I, II and III, as a new panel “c” in **Revised Fig. 4**.

We have added the following sentence in the revised manuscript: “*Of note, a large proportion of TFs were common between E10 and E11 in each cluster (48.6% in Cluster I, 33.4% in Cluster II and 30.3% in Cluster III) (Fig. 4c).*”

- *It would be great to see some validations whether indeed temporal changes in transcription factor genes are necessary for the AGM hematopoiesis.*

→ As mentioned above, we believe that the functional validation of the genes found differentially expressed is a tremendous amount of work and is therefore out of scope of the present study. Our goal is to provide unique and reliable data that will be useful to study the functional meaning of the temporal/spatial changes in transcription factor genes occurring during HSC fate acquisition.

- *It is intriguing that Gata2 expression is specifically enriched in nonHE, not in preHSC or EHT, in both E10 and E11, given the literature on Gata2 in HSCs.*

→ In fact, *Gata2* mRNA is also detected in other cell types but to a less extent than in endothelial cells (at both E11 and E10, **Supplementary data 8a and b**, respectively). The expression of *Gata2* in endothelial cells is known and was previously reported in various publications (Zhou *et al.* 2016, Swiers *et al.* 2013, Khandekar *et al.* 2007). A recent study has also shown the dynamic pulsatile expression of GATA2 in aortic cells occurring during their transition toward a hematopoietic fate (Eich *et al.* 2018). It therefore could explain the variability of *Gata2* expression in between cells and therefore in our scRNAseq data (compared to other studies).

- *Supplementary Figure 7 shows overlap between upregulated genes in IAHCs from RaceID cluster 1 and BM progenitors, and upregulated genes in IAHC cells from RaceID cluster 2 and BM HSCs. It would be informative if the analysis be done using the whole transcriptome, not upregulated, to that of BM and FL progenitors and HSCs.*

→ We performed the analysis by using the whole transcriptome as suggested by Reviewer 1. However, the comparison between two datasets generated by different methods is always delicate due to the variation in method sensitivity. Importantly, housekeeping genes are usually detected at very different levels, which results in batch effects that might not be related to the underlying biology.

We decided to remove the initial comparative analysis of our data with the one from Lara-Astiaso *et al.* (Science, 2014). It is replaced by the above comparative data analysis with the Zhou *et al.* paper (Nature, 2016), which is a more relevant comparative dataset to our study. The analysis is described in the text of the revised manuscript and data are depicted in a new **Supplementary Fig. 8**.

- *Currently, no signaling pathways are described in the transition from nonHE, HE, pre-HSC, and EHT. It would be good to see an analysis and discussion of the previously described signaling pathways reported functioning in the AGM hematopoiesis (for example, Notch, BMP, and cKIT) in relationship to some of the transcription factors such as Gata2. Additionally, Zhou et al. identified mTORC2 signaling to be highly enriched in pre-HSC-I compared to ECs (Nature, 2017). Perhaps, the authors can in silico analyze mTORC2-signaling pathway in their cell populations.*

→ As requested by Reviewer 1, we performed a KEGG pathway analysis on our E11 dataset. As expected, we identified several pathways to be active, including Notch, BMP,

VEGF, PDGF and TGF pathways (notably in the non HE/HE population, **Supplementary data 9 [highlighted in blue]**). More precisely to answer Reviewer 1 question, specific genes related to the BMP pathway (e.g. *Bmp4*, *Bmpr1a*, *Bmpr2*, *Smad 6*, *Smad 7*), Notch pathway (e.g. *Dll1*, *Dll4*, *Notch1*, *Notch 3*, *Notch 4*, *Jag1*) and mTORC2 pathway (e.g. *Rictor*, *mTor*, *Deptor*, *Tti1*) were expressed. We examined in more details the KEGG cytokine pathway (including the Kit pathway). We generated a heatmap where genes present in each cytokine pathway (according to KEGG analysis) were on the y-axis and E11 single cells (ordered along the pseudotime axis) were on the x-axis (**Supplementary data 10a**). It shows the dynamic expression of some known important regulators, including c-kit (Rybtsov *et al.* 2014) and IL-3 (through IL-3R) (Robin *et al.* 2006). Three groups of cytokines/growth factors were more active in endothelium, IAHC cells or committed progenitors. We performed a KEGG pathway analysis on these three datasets and found the involvement of interleukins, TGF, chemokines and/or BMP in the endothelium and IAHC cells (**Supplementary data 10b,c**). Pathways involving hemostasis and cell surface interactions at the vascular wall were active in the committed progenitor group, possibly reflecting the detachment of these cells to the circulation (**Supplementary data 10d**). Molecules involved in the immune system/inflammation were also active in the three groups (TNF, IL-6), in accordance to the literature where the inflammatory process was highlighted as an important key regulator of HSC development (Travnickova *et al.* 2015, Espín-Palazón *et al.* 2014, Li *et al.* 2014). The above-mentioned cytokine pathway analysis is described in the text of the revised manuscript and data are depicted in a new **Supplementary Fig. 11**.

We also added the following paragraph in the manuscript: “*To identify the pathways active in the aortic populations, we performed a KEGG pathway analysis on our E11 dataset. As expected, specific genes related to the BMP pathway (e.g. Bmp4, Bmpr1a, Bmpr2, Smad 6, Smad 7), Notch pathway (e.g. Dll1, Dll4, Notch1, Notch 3, Notch 4, Jag1) and mTORC2 pathway (e.g. Rictor, mTor, Deptor, Tti1) were expressed (data not shown). We then examined in more details the KEGG cytokine pathway. A heatmap was generated where genes present in each cytokine pathway (according to KEGG analysis) were on the y-axis and E11 single cells (ordered along the pseudotime axis) were on the x-axis (Supplementary Fig. 11a). It revealed the dynamic expression of some known important regulators, including c-kit⁵⁵ and IL-3 (through IL-3R)²⁹. Three groups of cytokines/growth factors were more active in endothelium, IAHC cells or committed progenitors. A KEGG pathway analysis performed on these three datasets highlighted the involvement of interleukins, TGF, chemokines and/or BMP in the endothelium and IAHC cells (Supplementary Fig. 11b,c). Pathways involving hemostasis and cell surface interactions at the vascular wall were active in the committed progenitor group, possibly reflecting the detachment of these cells to the circulation (Supplementary Fig. 11d). Molecules involved in the immune system/inflammation were also active in the three groups (i.e. TNF, IL-6), in accordance to the literature where the inflammatory process was recently reported as an important key regulator of HSC development⁵⁶. Our datasets will be useful to investigate in the future the requirement for specific cytokines and growth factors at specific time points during endothelial specification and HSC characteristic acquisition.*”

- Sugimura *et al.* recently used *ERG*, *HOXA5*, *HOXA9*, *HOXA10*, *LCOR*, *RUNX1* and *SPI1* to generate HSPCs from human PS cells (Nature, 2017). Lis *et al.* used *Fosb*, *Gfi1*, *Runx1*, and *Spi1* to reprogram endothelial cells to functional HSCs (Nature, 2017). Intriguingly, these factors seem to be expressed in a nondiscriminatory manner in nonHE, HE, pre-HSC, and progenitors. Thus, it is not so intuitive how the data from this manuscript can be useful for the future HSC generation or reprogramming.

→ We thank Reviewer 1 for this very important question. We looked in our E11 dataset at the expression of the factors used for reprogramming human iPSCs into hemogenic

endothelial cells and then into HSC-like cells (*Erg*, *Hoxa5*, *Hoxa9*, *Hoxa10*, *Lcor*, *Runx1* and *Spi1*) in Sugimura *et al.* paper as well as the factors used for reprogramming adult mouse endothelial cells into HSCs (*Fosb*, *Gfi1*, *Runx1* and *Spi1*) in Lis *et al.* paper (**Supplementary data 11a-c, g-i, m-o**). We also looked at the expression of these factors in the data of Zhou *et al.* (**Supplementary data 11d-f, j-l, p-r**). We found very similar patterns of expression in both studies. Indeed, (i) *Erg* and *Lcor* were not detected (**Supplementary data 11a,b,d,e**), (ii) *Fosb*, *Hoxa5*, *Hoxa9* and *Hoxa10* were very lowly expressed (**Supplementary data 11c,f, g-l**), and (iii) *Spi1*, *Runx1* and *Gfi1* were expressed as expected in EHT cells and most cluster cells (pre-HSCs and/or progenitors) in our data and in the Zhou populations (at the exception of the t1 CD201^{low} population, also as expected since we identified them as non-IAHC cells [see comments above]) (**Supplementary data 11m-r**). This analysis therefore shows that some of the factors used for reprogramming are found *in vivo*, differentially expressed in non-HE, HE, pre-HSC, and/or progenitors. The culture conditions developed to generate HSCs *in vitro* do not recreate a complete *in vivo* microenvironment but instead reproduces few of its elements (e.g. addition of cytokines or vascular-niche-derived angiocrine factors, activation/inhibition of specific pathways). Cell-cell contact is also an obvious regulatory element that is not (yet) optimally reproduced *in vitro*. Altogether, it could very well explain why some factors are needed *in vitro* but not *in vivo* (due to the redundant role of such factors with others *in vivo*). On the other hand, some factors that need to be discovered are obviously still missing to obtain the generation of HSCs that are fully functional and molecularly similar to *bona fide* HSCs. We believe that studying gene expression in the physiological context of the aorta is essential to understand the *in vivo* context of HSC production and can lead the way to improve the *in vitro* HSC production. In that perspective, our study should be useful. We added the following sentences in the discussion of the revised manuscript: “*Interestingly, we found a very similar expression pattern of the TFs used in the above-mentioned studies by looking at our transcriptional regulatory networks and the one from Zhou et al.*³². *As expected, Runx1, Spi1 and Gfi1 were expressed in EHT cells and most cluster cells (pre-HSCs and/or progenitors). However, some TFs were either weakly (Hoxa5, Hoxa9, Hoxa10, Fosb) or not (Erg, Lcor) detected. TF requirement might differ in vivo and in vitro because the culture conditions developed to generate HSCs in vitro do not recreate a complete in vivo microenvironment but instead reproduces few of its elements so far (e.g. addition of cytokines or vascular-niche-derived angiocrine factors, activation/inhibition of specific pathways)*^{2, 3}. *Shear stress and 3D cell-cell contact are also obvious regulatory elements that are not (yet) optimally reproduced in vitro. Therefore, some TFs might be needed in vitro but not in vivo due to a redundant role with other TFs in vivo. On the other hand, some factors that need to be discovered are obviously still missing to obtain the generation of HSCs that are fully functional and molecularly similar to bona fide HSCs. Studying gene expression in the physiological context of the aorta is thus essential to understand the in vivo context of HSC production and to lead the way to improve in vitro HSC production.*”

Supplementary data 1

Supplementary data 2

Supplementary data 3

Supplementary data 4

Supplementary data 5

Supplementary data 6

Supplementary data 7

Supplementary data 8

Supplementary data 9

Supplementary data 10

Supplementary data 11

Response to the Reviewer 2 comments

The molecular mechanisms that drive the emergence of HSCs from hemogenic endothelium has been to dissect because of the difficulty in profiling single cells. More generally, the spectrum of cellular heterogeneity as HSCs are specified and as they emerge has likewise been difficult to characterize for the same reason. In this manuscript Baron et al use scRNA-Seq to address these questions, and couple this with some smFISH validation, and they showcase a automated cell picking technology to compare dorsal versus ventral IAHCs transcriptomes. Overall, this is a solid contribution, and I have only minor suggestions as listed below.

→ We thank Reviewer 2 for his/her appreciation of our data and manuscript and his/her support. We replied to his/her minor comments and questions below (see below Supplementary data). We incorporated the most relevant data and findings in the revised version to improve the manuscript. We are of course open to add supplementary data if Reviewer 2 believes that it is necessary.

- *The tSNE plots do not agree with the RACE ID cluster assignments, and therefore the tSNE plots themselves, while popular, are somewhat confusion and perhaps not the most appropriate way to present the scRNA-Seq data.*

→ RaceID clustering and t-SNE are two distinct methods to cluster and visualize datasets. Indeed, there are often small discrepancies in the results of both analyses since the underlying algorithms that generate the clustering and the t-SNE results are different. The fact that both t-SNE and clustering largely overlap indicates that clustering is robust and does not solely depend on the algorithm that is used.

- *Figure 1: The authors show scRNA-Seq of Kit, but what about the concordance between marker expression and RNA for the others (CD45 and CD31)?*

→ As requested by Reviewer 2, we provide the t-SNE maps showing *CD45* (*Ptprc*) and *CD31* (*Pecam1*) transcript expression (**Supplementary data 1a and 1b**, respectively). Similar to *Kit* expression, the cells have variable levels of expression of *Ptprc* and *Pecam1*. We did not include the t-SNE maps in the revised manuscript but we added the following sentence in the revised manuscript: “Cells had varying levels of *Kit*, *Pecam1* (*CD31*) and *Ptprc* (*CD45*) mRNA expression (**Fig. 1c**, and data not shown).”

- *in silico purification of IAHC: what threshold used to define Gpr56+? Some non-IAHC cells appear to be positive in Figure 1.*

→ To *in silico* purify IAHC cells, we selected the cells that had more than one *Gpr56* transcript and filtered out the cells that had more than two transcripts of one or more of the following non-IAHC genes, *Cxcl12*, *Prrx1*, *Epha7* and/or *Pdgfrb*. The *in silico* purified cells used for the study (depicted in red in **Supplementary data 2a**) are pure IAHC cells, with no contamination by non-IAHC cells. We agree with Reviewer 2 that some (although very few) *Gpr56*⁺ cells seemed to be present in the non-IAHC population (in RaceID clusters 1, 2, 3 as shown in Fig.1b). However, these few cells expressed lower levels of *Gpr56* transcripts when compared to the true IAHC cells (in RaceID clusters 4 and 5 as shown in Fig.1b) and high levels of non-IAHC markers (Supplementary Fig. 2c-j). They were therefore considered as non-IAHC cells and excluded for further analysis. As shown in **Supplementary data 2b**, the expression of several known genes confirmed the identity of the populations that we identified as IAHC and non-IAHC populations.

We have included the Supplementary data 2a as the new panel “f” in revised **Fig. 1** and the following sentence in the revised manuscript: “However, scRNA-seq represents a powerful tool to correct for cell contamination by *in silico* purifying IAHC cells based on specific marker gene expression. To study pure IAHC cells, we therefore selected the cells that had

more than one *Gpr56* transcript and filtered out the cells that had more than two transcripts of one or more of the non-IAHC genes (Fig. 1f).”

- Pg4: "scRNA-seq represents a powerful tool to correct for cell contamination by *in silico* purifying IAHC cells based on the expression of known IAHC marker genes (such as *Gpr56*" <-- This claim depends on the sensitivity of the scRNA-Seq. If there are libraries in which *Gpr56* has 'dropped out', then this method will falsely assign the cell to a non-IAHC group. What is the CEL-Seq sensitivity? Could you please reply to this question? Can this issue of drop out be addressed to some extent by using drop out correction methods such as MAGIC?

→ By using single-molecule RNA-FISH (smRNA-FISH) data, the sensitivity of CEL-Seq was analyzed (Grün *et al.* Nature Methods, 2014). While mean expression correlated strongly (between both methods), the smRNA-FISH sensitivity was eightfold higher. Assuming 100% sensitivity of smRNA-FISH yields an estimated CEL-Seq sensitivity of 12.5%. However, the spike-in counts suggested an efficiency of 3.4%, which might be explained by various biological and technical variables (e.g. RNA degradation prior to amplification). Because the actual sensitivity of smRNA-FISH has been shown to be >80%, the true sensitivity of our CEL-Seq experiments was estimated at 10%.

→ The expression of *Gpr56* is a very good indication that the cells are IAHC cells (as confirmed by smRNA-FISH in this manuscript [Fig.2g-i] and by ISH in Soleimani *et al.* (2015). If there are libraries in which *Gpr56* has dropped out, the non-IAHC cell markers (such as *Cxcl12* or *Col3a1*) can be used to exclude non-IAHC cells and therefore to enrich the IAHC cell population (which does not express these marker genes).

As suggested by Reviewer 2, we used the MAGIC algorithm to address the potential *Gpr56* dropout issue. We generated two histograms representing the number of *Gpr56* transcripts per cell with and without dropout correction using MAGIC (van Dijk D. *et al.*, bioRxiv, 2017). The frequency of cells with no *Gpr56* transcripts (corresponding to the non-IAHC cells, grey column) does not change with or without MAGIC correction (Supplementary data 3a, b, respectively), indicating that very few *Gpr56*⁺ cells were dropped out. Of note, we nicely see that after MAGIC dropout correction the histogram shows a bimodal distribution that better segregates *Gpr56*⁺ and *Gpr56*⁻ cells. Whereas this does not influence our IAHC cell *in silico* selection, it emphasizes the capacity of MAGIC to structure single cell RNA-Seq data. We also generated a t-SNE map where the red dots are the *in silico* purified IAHC cells without MAGIC dropout correction and where the red dots circled in blue are the *in silico* purified IAHC cells after MAGIC dropout correction (Supplementary data 3c). Since most red cells are circled in blue, it again shows that very few *Gpr56*⁺ cells are dropped out in our analysis.

- RACE ID cluster labels in Figure 2D are not clear. There are several points that have IDs like 13, 14 etc. Are these just cells in close proximity by different RACE ID clusters? And there are even cells labeled as cluster 7, but the legend only goes up to 4. Related to this, how are 'main' clusters identified (Pg6)? Is this just a call made by eye?

→ We apologize for the mistake. The graph was correct in Fig. 1b but the wrong graphs were included in the initial Fig. 2d and 3c. The correct graphs, which show only the main RaceID clusters and not the outliers, are now included in revised Fig. 2d and revised Fig. 3c. The color coding being changed, corrections have been made accordingly in revised Fig. 5h,i and revised Fig. 6e,f,h,i.

- Figure 3A: The EHT cluster appears to be detected as a separate branch by Monocle. Can the authors explain this?

→ The backbone (black line) shown in the Monocle plot in Fig. 3a represents the shortest path from the endothelial cells to the progenitors. We agree with Reviewer 2 that the EHT population appeared slightly outside of this trajectory. However, Monocle collapses all

side branches onto the backbone, which ensured that all EHT cells were projected on the backbone at the right point in pseudotime. Importantly, the backbone runs through the section of the plot where HE and EHT cells mix and only a portion of EHT cells project on the side branch.

- *What units are the mean columns in the supp tables? The values are quite large.*

→ The column means are the transcript numbers, which indeed can be very high for some highly expressed genes.

- *Are there any hints of genes/signaling pathways related to dorsal ventral patterning in the D vs V comparisons?*

→ We could not find genes or signaling pathways specifically related to the dorsal ventral patterning in our dataset, which reinforces the transcriptome similarity of the ventral and dorsal IAHCs. Our data support the fact that both ventral and dorsal IAHCs contain progenitors and pre-HSCs (Taoudi *et al.* 2007, Souilhol *et al.* 2016).

- *The authors claim "It will pave the way for improving HSC production in vitro". But the authors do not discuss any of the new transcription factors that emerge from their pseudotime analysis. What new knowledge has been gained here? Performing an GSEA-style analysis of signaling pathways that correlate with the pseudotime analysis would, in principle, be helpful to propose new means to coax iPSC-HE to HSCs.*

→ We thank Reviewer 2 for this important point. We performed a KEGG pathway analysis on our E11 dataset. As expected, we identified several pathways to be active, including Notch, BMP, VEGF, PDGF and TGF pathways (notably in the non HE/HE population, **Supplementary data 4 [highlighted in blue]**). More precisely to answer Reviewer 1 question, specific genes related to the BMP pathway (e.g. *Bmp4*, *Bmpr1a*, *Bmpr2*, *Smad 6*, *Smad 7*), Notch pathway (e.g. *Dll1*, *Dll4*, *Notch1*, *Notch 3*, *Notch 4*, *Jag1*) and mTORC2 pathway (e.g. *Rictor*, *mTor*, *Deptor*, *Tti1*) were expressed. We examined in more details the KEGG cytokine pathway (including the Kit pathway). We generated a heatmap where genes present in each cytokine pathway (according to KEGG analysis) were on the y-axis and E11 single cells (ordered along the pseudotime axis) were on the x-axis (**Supplementary data 5a**). It shows the dynamic expression of some known important regulators, including c-kit (Rybtsov *et al.* 2014) and IL-3 (through IL-3R) (Robin *et al.* 2006). Three groups of cytokines/growth factors were more active in endothelium, IAHC cells or committed progenitors. We performed a KEGG pathway analysis on these three datasets and found the involvement of interleukins, TGF, chemokines and/or BMP in the endothelium and IAHC cells (**Supplementary data 5b,c**). Pathways involving hemostasis and cell surface interactions at the vascular wall were active in the committed progenitor group, possibly reflecting the detachment of these cells to the circulation (**Supplementary data 5d**). Molecules involved in the immune system/inflammation were also active in the three groups (TNF, IL-6), in accordance to the literature where the inflammatory process was highlighted as an important key regulator of HSC development (Travnickova *et al.* 2015, Espín-Palazón *et al.* 2014, Li *et al.* 2014). The above-mentioned cytokine pathway analysis is described in the text of the revised manuscript and data are depicted in a new **Supplementary Fig. 11**.

We also added the following paragraph in the manuscript: “*To identify the pathways active in the aortic populations, we performed a KEGG pathway analysis on our E11 dataset. As expected, specific genes related to the BMP pathway (e.g. Bmp4, Bmpr1a, Bmpr2, Smad 6, Smad 7), Notch pathway (e.g. Dll1, Dll4, Notch1, Notch 3, Notch 4, Jag1) and mTORC2 pathway (e.g. Rictor, mTor, Deptor, Tti1) were expressed (data not shown). We then examined in more details the KEGG cytokine pathway. A heatmap was generated where genes present in each cytokine pathway (according to KEGG analysis) were on the y-axis and*

E11 single cells (ordered along the pseudotime axis) were on the x-axis (**Supplementary Fig. 11a**). It revealed the dynamic expression of some known important regulators, including *c-kit*⁵⁵ and IL-3 (through IL-3R)²⁹. Three groups of cytokines/growth factors were more active in endothelium, IAHC cells or committed progenitors. A KEGG pathway analysis performed on these three datasets highlighted the involvement of interleukins, TGF, chemokines and/or BMP in the endothelium and IAHC cells (**Supplementary Fig. 11b,c**). Pathways involving hemostasis and cell surface interactions at the vascular wall were active in the committed progenitor group, possibly reflecting the detachment of these cells to the circulation (**Supplementary Fig. 11d**). Molecules involved in the immune system/inflammation were also active in the three groups (i.e. TNF, IL-6), in accordance to the literature where the inflammatory process was recently reported as an important key regulator of HSC development (Espín-Palazón *et al.* 2017). Our datasets will be useful to investigate in the future the requirement for specific cytokines and growth factors at specific time points during endothelial specification and HSC characteristic acquisition.”

We also looked in our E11 dataset at the expression of the factors used for reprogramming human iPSCs into hemogenic endothelial cells and then into HSC-like cells (*Erg*, *Hoxa5*, *Hoxa9*, *Hoxa10*, *Lcor*, *Runx1* and *Spi1*) in Sugimura *et al.* paper as well as the factors used for reprogramming adult mouse endothelial cells into HSCs (*Fosb*, *Gfi1*, *Runx1* and *Spi1*) in Lis *et al.* paper (**Supplementary data 6a-c, g-i, m-o**). We also looked at the expression of these factors in the data of Zhou *et al.* (**Supplementary data 6d-f, j-l, p-r**). We found very similar patterns of expression in both studies. Indeed, (i) *Erg* and *Lcor* were not detected (**Supplementary data 6a,b,d,e**), (ii) *Fosb*, *Hoxa5*, *Hoxa9* and *Hoxa10* were very lowly expressed (**Supplementary data 6c,f, g-l**), and (iii) *Spi1*, *Runx1* and *Gfi1* were expressed as expected in EHT cells and most cluster cells (pre-HSCs and/or progenitors) in our data and in the Zhou populations (at the exception of the t1 CD201^{low} population, also as expected since we identified them as non-IAHC cells [see comments above]) (**Supplementary data 6m-r**). This analysis therefore shows that some of the factors used for reprogramming are found *in vivo*, differentially expressed in non-HE, HE, pre-HSC, and/or progenitors. The culture conditions developed to generate HSCs *in vitro* do not recreate a complete *in vivo* microenvironment but instead reproduces few of its elements (e.g. addition of cytokines or vascular-niche-derived angiocrine factors, activation/inhibition of specific pathways). Cell-cell contact is also an obvious regulatory element that is not (yet) optimally reproduced *in vitro*. Altogether, it could very well explain why some factors are needed *in vitro* but not *in vivo* (due to the redundant role of such factors with others *in vivo*). On the other hand, some factors that need to be discovered are obviously still missing to obtain the generation of HSCs that are fully functional and molecularly similar to *bona fide* HSCs. We believe that studying gene expression in the physiological context of the aorta is essential to understand the *in vivo* context of HSC production and can lead the way to improve the *in vitro* HSC production. In that perspective, our study should be useful.

We added the following sentences in the discussion of the revised manuscript: “Interestingly, we found a very similar expression pattern of the TFs used in the above-mentioned studies by looking at our transcriptional regulatory networks and the one from Zhou *et al.*³². As expected, *Runx1*, *Spi1* and *Gfi1* were expressed in EHT cells and most cluster cells (pre-HSCs and/or progenitors). However, some TFs were either weakly (*Hoxa5*, *Hoxa9*, *Hoxa10*, *Fosb*) or not (*Erg*, *Lcor*) detected. TF requirement might differ *in vivo* and *in vitro* because the culture conditions developed to generate HSCs *in vitro* do not recreate a complete *in vivo* microenvironment but instead reproduces few of its elements so far (e.g. addition of cytokines or vascular-niche-derived angiocrine factors, activation/inhibition of specific pathways)^{2, 3}. Shear stress and 3D cell-cell contact are also obvious regulatory elements that are not (yet) optimally reproduced *in vitro*. Therefore, some TFs might be needed *in vitro* but not *in vivo*

due to a redundant role with other TFs in vivo. On the other hand, some factors that need to be discovered are obviously still missing to obtain the generation of HSCs that are fully functional and molecularly similar to bona fide HSCs. Studying gene expression in the physiological context of the aorta is thus essential to understand the in vivo context of HSC production and to lead the way to improve in vitro HSC production.”

Supplementary data 1

Supplementary data 2

a

b

Supplementary data 3

Supplementary data 4

Supplementary data 5

Supplementary data 6

REVIEWERS' COMMENTS:

Reviewer #1 (Remarks to the Author):

The authors responded appropriately to all the comments this reviewer raised. I am happy to recommend this paper for publication to Nature Communications.

Reviewer #2 (Remarks to the Author):

The authors have addressed all of my questions.